# Distributed Variational Inference in Sparse Gaussian Process Regression and Latent Variable Models

**Yarin Gal**[*]          **Mark van der Wilk**[*]          **Carl E. Rasmussen**

University of Cambridge
{yg279,mv310,cer54}@cam.ac.uk

## Abstract

Gaussian processes (GPs) are a powerful tool for probabilistic inference over functions. They have been applied to both regression and non-linear dimensionality reduction, and offer desirable properties such as uncertainty estimates, robustness to over-fitting, and principled ways for tuning hyper-parameters. However the scalability of these models to big datasets remains an active topic of research.

We introduce a novel re-parametrisation of variational inference for sparse GP regression and latent variable models that allows for an efficient distributed algorithm. This is done by exploiting the decoupling of the data given the inducing points to re-formulate the evidence lower bound in a Map-Reduce setting.

We show that the inference scales well with data and computational resources, while preserving a balanced distribution of the load among the nodes. We further demonstrate the utility in scaling Gaussian processes to big data. We show that GP performance improves with increasing amounts of data in regression (on flight data with 2 million records) and latent variable modelling (on MNIST). The results show that GPs perform better than many common models often used for big data.

## 1   Introduction

Gaussian processes have been shown to be flexible models that are able to capture complicated structure, without succumbing to over-fitting. Sparse Gaussian process (GP) regression [Titsias, 2009] and the Bayesian Gaussian process latent variable model (GPLVM, Titsias and Lawrence [2010]) have been applied in many tasks, such as regression, density estimation, data imputation, and dimensionality reduction. However, the use of these models with big datasets has been limited by the scalability of the inference. For example, the use of the GPLVM with big datasets such as the ones used in continuous-space natural language disambiguation is quite cumbersome and challenging, and thus the model has largely been ignored in such communities.

It is desirable to scale the models up to be able to handle large amounts of data. One approach is to spread computation across many nodes in a distributed implementation. Brockwell [2006]; Wilkinson [2005]; Asuncion et al. [2008], among others, have reasoned about the requirements such distributed algorithms should satisfy. The inference procedure should:

1. distribute the computational load evenly across nodes,
2. scale favourably with the number of nodes,
3. and have low overhead in the global steps.

In this paper we scale sparse GP regression and latent variable modelling, presenting the first distributed inference algorithm for the models able to process datasets with millions of points. We derive a re-parametrisation of the variational inference proposed by Titsias [2009] and Titsias and Lawrence [2010], unifying the two, which allows us to perform inference using the original guarantees. This is achieved through the fact that conditioned on the inducing inputs, the data decouples and the variational parameters can be updated independently on different nodes, with the only communi-

---

[*]Authors contributed equally to this work.

cation between nodes requiring constant time. This also allows the optimisation of the embeddings in the GPLVM to be done in parallel.

We experimentally study the properties of the suggested inference showing that the inference scales well with data and computational resources, and showing that the inference running time scales inversely with computational power. We further demonstrate the practicality of the inference, inspecting load distribution over the nodes and comparing run-times to sequential implementations.

We demonstrate the utility in scaling Gaussian processes to big data showing that GP performance improves with increasing amounts of data. We run regression experiments on 2008 US flight data with 2 million records and perform classification tests on MNIST using the latent variable model. We show that GPs perform better than many common models which are often used for big data.

The proposed inference was implemented in Python using the Map-Reduce framework [Dean and Ghemawat, 2008] to work on multi-core architectures, and is available as an open-source package[1]. The full derivation of the inference is given in the supplementary material as well as additional experimental results (such as robustness tests to node failure by dropping out nodes at random). The open source software package contains an extensively documented implementation of the derivations, with references to the equations presented in the supplementary material for explanation.

## 2 Related Work

Recent research carried out by Hensman et al. [2013] proposed stochastic variational inference (SVI, Hoffman et al. [2013]) to scale up sparse Gaussian process regression. Their method trained a Gaussian process using mini-batches, which allowed them to successfully learn from a dataset containing 700,000 points. Hensman et al. [2013] also note the applicability of SVI to GPLVMs and suggest that SVI for GP regression can be carried out in parallel. However SVI also has some undesirable properties. The variational marginal likelihood bound is less tight than the one proposed in Titsias [2009]. This is a consequence of representing the variational distribution over the inducing targets $q(\mathbf{u})$ explicitly, instead of analytically deriving and marginalising the optimal form. Additionally SVI needs to explicitly optimise over $q(\mathbf{u})$, which is not necessary when using the analytic optimal form. The noisy gradients produced by SVI also complicate optimisation; the inducing inputs need to be fixed in advance because of their strong correlation with the inducing targets, and additional optimiser-specific parameters, such as step-length, have to be introduced and fine-tuned by hand. Heuristics do exist, but these points can make SVI rather hard to work with.

Our approach results in the same lower bound as presented in Titsias [2009], which averts the difficulties with the approach above, and enables us to scale GPLVMs as well.

## 3 The Gaussian Process Latent Variable Model and Sparse GP Regression

We now briefly review the sparse Gaussian process regression model [Titsias, 2009] and the Gaussian process latent variable model (GPLVM) [Lawrence, 2005; Titsias and Lawrence, 2010], in terms of model structure and inference.

### 3.1 Sparse Gaussian Process Regression

We consider the standard Gaussian process regression setting, where we aim to predict the output of some unknown function at new input locations, given a training set of $n$ inputs $\{X_1, \ldots, X_n\}$ and corresponding observations $\{Y_1, \ldots, Y_n\}$. The observations consist of the latent function values $\{F_1, \ldots, F_n\}$ corrupted by some i.i.d. Gaussian noise with precision $\beta$. This gives the following generative model[2]:

$$F(X_i) \sim \mathcal{GP}(0, k(X, X)), \quad Y_i \sim \mathcal{N}(F_i, \beta^{-1}I)$$

For convenience, we collect the data in a matrix and denote single data points by subscripts.

$$X \in \mathbb{R}^{n \times q}, \quad F \in \mathbb{R}^{n \times d}, \quad Y \in \mathbb{R}^{n \times d}$$

We can marginalise out the latent $F$ analytically in order to find the predictive distribution and marginal likelihood. However, this consists of an inversion of an $n \times n$ matrix, thus requiring $\mathcal{O}(n^3)$ time complexity, which is prohibitive for large datasets.

To address this problem, many approximations have been developed which aim to summarise the behaviour of the regression function using a sparse set of $m$ input-output pairs, instead of the entire dataset[3]. These input-output pairs are termed "inducing points" and are taken to be sufficient statistics for any predictions. Given the inducing inputs $Z \in \mathbb{R}^{m \times q}$ and targets $\mathbf{u} \in \mathbb{R}^{m \times d}$, predictions can be made in $\mathcal{O}(m^3)$ time complexity:

$$p(F^*|X^*, Y) \approx \int \mathcal{N}\left(F^*; k_{*m} K_{mm}^{-1}\mathbf{u}, k_{**} - k_{*m} K_{mm}^{-1} k_{m*}\right) p(\mathbf{u}|Y, X) \mathrm{d}\mathbf{u} \qquad (3.1)$$

where $K_{mm}$ is the covariance between the $m$ inducing inputs, and likewise for the other subscripts.

Learning the function corresponds to inferring the posterior distribution over the inducing targets $\mathbf{u}$. Predictions are then made by marginalising $\mathbf{u}$ out of equation 3.1. Efficiently learning the posterior over $\mathbf{u}$ requires an additional assumption to be made about the relationship between the training data and the inducing points, such as a deterministic link using only the conditional GP mean $F = K_{nm} K_{mm}^{-1}\mathbf{u}$. This results in an overall computational complexity of $\mathcal{O}(nm^2)$.

Quiñonero-Candela and Rasmussen [2005] view this procedure as changing the prior to make inference more tractable, with $Z$ as hyperparameters which can be tuned using optimisation. However, modifying the prior in response to training data has led to over-fitting. An alternative sparse approximation was introduced by Titsias [2009]. Here a variational distribution over $\mathbf{u}$ is introduced, with $Z$ as variational parameters which tighten the corresponding evidence lower bound. This greatly reduces over-fitting, while retaining the improved computational complexity. It is this approximation which we further develop in this paper to give a distributed inference algorithm. A detailed derivation is given in section 3 of the supplementary material.

### 3.2   Gaussian Process Latent Variable Models

The Gaussian process latent variable model (GPLVM) can be seen as an *unsupervised* version of the regression problem above. We aim to infer both the inputs, which are now latent, and the function mapping at the same time. This can be viewed as a non-linear generalisation of PCA [Lawrence, 2005]. The model set-up is identical to the regression case, only with a prior over the latents $X$.

$$X_i \sim \mathcal{N}(X_i; 0, I), \quad F(X_i) \sim \mathcal{GP}(0, k(X, X)), \quad Y_i \sim \mathcal{N}(F_i, \beta^{-1} I)$$

A Variational Bayes approximation for this model has been developed by Titsias and Lawrence [2010] using similar techniques as for variational sparse GPs. In fact, the sparse GP can be seen as a special case of the GPLVM where the inputs are given zero variance. The main task in deriving approximate inference revolves around finding a variational lower bound to:

$$p(Y) = \int p(Y|F) p(F|X) p(X) \mathrm{d}(F, X)$$

Which leads to a Gaussian approximation to the posterior $q(X) \approx p(X|Y)$, explained in detail in section 4 of the supplementary material. In the next section we derive a distributed inference scheme for both models following a re-parametrisation of the derivations of Titsias [2009].

## 4   Distributed Inference

We now exploit the conditional independence of the data given the inducing points to derive a distributed inference scheme for both the sparse GP model and the GPLVM, which will allow us to easily scale these models to large datasets. The key equations are given below, with an in-depth explanation given in sections sections 3 and 4 of the supplementary material. We present a unifying derivation of the inference procedures for both the regression case and the latent variable modelling (LVM) case, by identifying that the explicit inputs in the regression case are identical to the latent inputs in the LVM case when their mean is set to the observed inputs and used with variance 0 (i.e. the latent inputs are fixed and not optimised).

We start with the general expression for the log marginal likelihood of the sparse GP regression model, after introducing the inducing points,

$$\log p(Y|X) = \log \int p(Y|F)p(F|X,\mathbf{u})p(\mathbf{u})\mathrm{d}(\mathbf{u},F).$$

The LVM derivation encapsulates this expression by multiplying with the prior over $X$ and then marginalising over $X$:

$$\log p(Y) = \log \int p(Y|F)p(F|X,\mathbf{u})p(\mathbf{u})p(X)\mathrm{d}(\mathbf{u},F,X).$$

We then introduce a free-form variational distribution $q(\mathbf{u})$ over the inducing points, and another over $X$ (where in the regression case, $p(X)$'s and $q(X)$'s variance is set to 0 and their mean set to $X$). Using Jensen's inequality we get the following lower bound:

$$\log p(Y|X) \geq \int p(F|X,\mathbf{u})q(\mathbf{u})\log \frac{p(Y|F)p(\mathbf{u})}{q(\mathbf{u})}\mathrm{d}(\mathbf{u},F)$$

$$= \int q(\mathbf{u})\left(\int p(F|X,\mathbf{u})\log p(Y|F)\mathrm{d}(F) + \log \frac{p(\mathbf{u})}{q(\mathbf{u})}\right)\mathrm{d}(\mathbf{u}) \qquad (4.1)$$

all distributions that involve $\mathbf{u}$ also depend on $Z$ which we have omitted for brevity. Next we integrate $p(Y)$ over $X$ to be able to use 4.1,

$$\log p(Y) = \log \int q(X)\frac{p(Y|X)p(X)}{q(X)}\mathrm{d}(X) \geq \int q(X)\left(\log p(Y|X) + \log \frac{p(X)}{q(X)}\right)\mathrm{d}(X) \qquad (4.2)$$

and obtain a bound which can be used for both models. Up to here the derivation is identical to the two derivations given in [Titsias and Lawrence, 2010; Titsias, 2009]. However, now we exploit the conditional independence given $\mathbf{u}$ to break the inference into small independent components.

## 4.1 Decoupling the Data Conditioned on the Inducing Points

The introduction of the inducing points *decouples the function values from each other* in the following sense. If we represent $Y$ as the individual data points $(Y_1; Y_2; ...; Y_n)$ with $Y_i \in \mathbb{R}^{1 \times d}$ and similarly for $F$, we can write the lower bound as a sum over the data points, since $Y_i$ are independent of $F_j$ for $j \neq i$:

$$\int p(F|X,\mathbf{u})\log p(Y|F)\mathrm{d}(F) = \int p(F|X,\mathbf{u})\sum_{i=1}^{n}\log p(Y_i|F_i)\mathrm{d}(F)$$

$$= \sum_{i=1}^{n}\int p(F_i|X_i,\mathbf{u})\log p(Y_i|F_i)\mathrm{d}(F_i)$$

Simplifying this expression and integrating over $X$ we get that each term is given by

$$-\frac{d}{2}\log(2\pi\beta^{-1}) - \frac{\beta}{2}\left(Y_iY_i^T - 2\langle F_i\rangle_{p(F_i|X_i,\mathbf{u})q(X_i)}Y_i^T + \langle F_iF_i^T\rangle_{p(F_i|X_i,\mathbf{u})q(X_i)}\right)$$

where we use triangular brackets $\langle F\rangle_{p(F)}$ to denote the expectation of $F$ with respect to the distribution $p(F)$.

Now, using calculus of variations we can find optimal $q(\mathbf{u})$ analytically. Plugging the optimal distribution into eq. 4.1 and using further algebraic manipulations we obtain the following lower bound:

$$\log p(Y) \geq -\frac{nd}{2}\log 2\pi + \frac{nd}{2}\log \beta + \frac{d}{2}\log |K_{mm}| - \frac{d}{2}\log |K_{mm} + \beta D|$$

$$-\frac{\beta}{2}A - \frac{\beta d}{2}B + \frac{\beta d}{2}Tr(K_{mm}^{-1}D) + \frac{\beta^2}{2}Tr(C^T \cdot (K_{mm} + \beta D)^{-1} \cdot C) - KL \qquad (4.3)$$

where

$$A = \sum_{i=1}^{n}Y_iY_i^T, \quad B = \sum_{i=1}^{n}\langle K_{ii}\rangle_{q(X_i)}, \quad C = \sum_{i=1}^{n}\langle K_{mi}\rangle_{q(X_i)}Y_i, \quad D = \sum_{i=1}^{n}\langle K_{mi}K_{im}\rangle_{q(X_i)}$$

and

$$KL = \sum_{i=1}^{n}KL(q(X_i)||p(X_i))$$

when the inputs are latent or set to 0 when they are observed.

Notice that the obtained unifying bound is identical to the ones derived in [Titsias, 2009] for the regression case and [Titsias and Lawrence, 2010] for the LVM case since $\langle K_{mi} \rangle_{q(X_i)} = K_{mi}$ for $q(X_i)$ with variance 0 and mean $X_i$. However, the terms are re-parametrised as independent sums over the input points – sums that can be computed on different nodes in a network without inter-communication. An in-depth explanation of the different transitions is given in the supplementary material sections 3 and 4.

## 4.2 Distributed Inference Algorithm

A parallel inference algorithm can be easily derived based on this factorisation. Using the Map-Reduce framework [Dean and Ghemawat, 2008] we can maintain different subsets of the inputs and their corresponding outputs on each node in a parallel implementation and distribute the global parameters (such as the kernel hyper-parameters and the inducing inputs) to the nodes, collecting only the partial terms calculated on each node.

We denote by $\mathbf{G}$ the set of global parameters over which we need to perform optimisation. These include $Z$ (the inducing inputs), $\beta$ (the observation noise), and $\mathbf{k}$ (the set of kernel hyper-parameters). Additionally we denote by $\mathbf{L}_k$ the set of local parameters on each node $k$ that need to be optimised. These include the mean and variance for each input point for the LVM model. First, we send to all end-point nodes the global parameters $\mathbf{G}$ for them to calculate the partial terms $\langle K_{mi} \rangle_{q(X_i)} Y_i$, $\langle K_{mi} K_{im} \rangle_{q(X_i)}$, $\langle K_{ii} \rangle_{q(X_i)}$, $Y_i Y_i^T$, and $KL(q(X_i)||p(X_i))$. The calculation of these terms is explained in more detail in the supplementary material section 4. The end-point nodes return these partial terms to the central node (these are $m \times m \times q$ matrices – constant space complexity for fixed $m$). The central node then sends the accumulated terms and partial derivatives back to the nodes and performs global optimisation over $\mathbf{G}$. In the case of the GPLVM, the nodes then concurrently perform local optimisation on $\mathbf{L}_k$, the embedding posterior parameters. In total, we have two Map-Reduce steps between the central node and the end-point nodes to follow:

1. The central node distributes $\mathbf{G}$,
2. Each end-point node $k$ returns a partial sum of the terms $A, B, C, D$ and $KL$ based on $\mathbf{L}_k$,
3. The central node calculates $\mathcal{F}, \partial\mathcal{F}$ ($m \times m \times q$ matrices) and distributes to the end-point nodes,
4. The central node optimises $\mathbf{G}$; at the same time the end-point nodes optimise $\mathbf{L}_k$.

When performing regression, the third step and the second part of the fourth step are not required. The appendices of the supplementary material contain the derivations of all the partial derivatives required for optimisation.

Optimisation of the global parameters can be done using any procedure that utilises the calculated partial derivative (such as scaled conjugate gradient [Møller, 1993]), and the optimisation of the local variables can be carried out by parallelising SCG or using local gradient descent. We now explore the developed inference empirically and evaluate its properties on a range of tasks.

## 5 Experimental Evaluation

We now demonstrate that the proposed inference meets the criteria set out in the introduction. We assess the inference on its scalability with increased computational power for a fixed problem size (strong scaling) as well as with proportionally increasing data (weak scaling) and compare to existing inference. We further explore the distribution of the load over the different nodes, which is a major inhibitor in large scale distributed systems.

In the following experiments we used a squared exponential ARD kernel over the latent space to automatically determine the dimensionality of the space, as in Titsias and Lawrence [2010]. We initialise our latent points using PCA and our inducing inputs using k-means with added noise. We optimise using both L-BFGS and scaled conjugate gradient [Møller, 1993].

### 5.1 Scaling with Computation Power

We investigate how much inference on a given dataset can be sped up using the proposed algorithm given more computational resources. We assess the improvement of the running time of the algo-

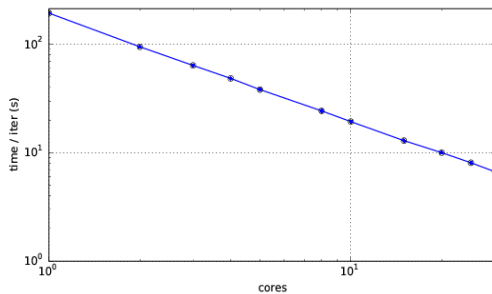

Figure 1: **Running time per iteration for 100K points synthetic dataset,** as a function of available cores on *log-scale*.

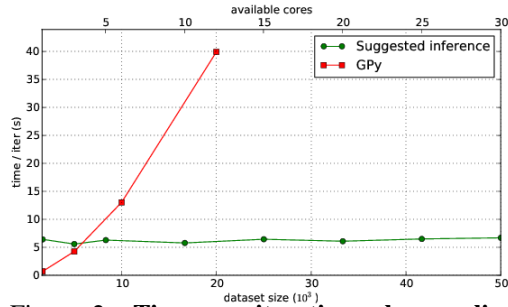

Figure 2: **Time per iteration when scaling the computational resources proportionally to dataset size up to 50K points.** Also shown standard inference (GPy) for comparison.

rithm on a synthetic dataset of which large amounts of data could easily be generated. The dataset was obtained by simulating a 1D latent space and transforming this non-linearly into 3D observations. 100K points were generated and the algorithm was run using an increasing number of cores and a 2D latent space. We measured the total running time the algorithm spent in each iteration.

Figure 1 shows the improvement of run-time as a function of available cores. We obtain a relation very close to the ideal $t \propto c \cdot (cores)^{-1}$. When doubling the number of cores from 5 to 10 we achieve a factor 1.93 decrease in computation time – very close to ideal. In a higher range, a doubling from 15 to 30 cores improves the running time by a factor of 1.90, so there is very little sign of diminishing returns. It is interesting to note that we observed a minuscule overhead of about 0.05 seconds per iteration in the global steps. This is due to the $m \times m$ matrix inversion carried out in each global step, which amounts to an additional time complexity of $\mathcal{O}(m^3)$ – constant for fixed $m$.

## 5.2 Scaling with Data and Comparison to Standard Inference

Using the same setup, we assessed the scaling of the running time as we increased both the dataset size and computational resources equally. For a doubling of data, we doubled the number of available CPUs. In the ideal case of an algorithm with only distributable components, computation time should be constant. Again, we measure the total running time of the algorithm per iteration. Figure 2 shows that we are able to effectively utilise the extra computational resources. Our total running time takes $4.3\%$ longer for a dataset scaled by 30 times.

Comparing the computation time to the standard inference scheme we see a significant improvement in performance in terms of running time. We compared to the sequential but highly optimised GPy implementation (see figure 2). The suggested inference significantly outperforms GPy in terms of running time given more computational resources. Our parallel inference allows us to run sparse GPs and the GPLVM on datasets which would simply take too long to run with standard inference.

## 5.3 Distribution of the Load

The development of parallel inference procedures is an active field of research for Bayesian nonparametric models [Lovell et al., 2012; Williamson et al., 2013]. However, it is important to study

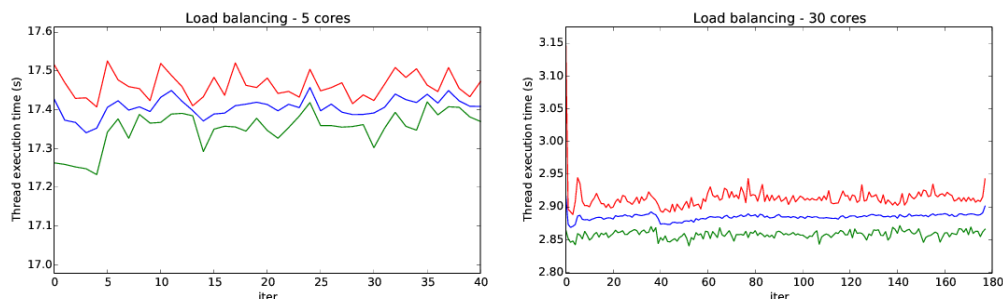

Figure 3: **Load distribution for each iteration.** The maximum time spent in a node is the rate limiting step. Shown are the minimum, mean and maximum execution times of all nodes when using 5 (left) and 30 (right) cores.

| Dataset | Mean | Linear | Ridge | RF | SVI 100 | SVI 200 | Dist GP 100 |
|---|---|---|---|---|---|---|---|
| Flight 7K | 36.62 | 34.97 | 35.05 | 34.78 | NA | NA | **33.56** |
| Flight 70K | 36.61 | 34.94 | 34.98 | 34.88 | NA | NA | **33.11** |
| Flight 700K | 36.61 | 34.94 | 34.95 | 34.96 | 33.20 | 33.00 | **32.95** |

Table 1: RMSE of flight delay (measured in minutes) for regression over flight data with 7K-700K points by predicting mean, linear regression, ridge regression, random forest regression (RF), Stochastic Variational Inference (SVI) GP regression with 100 and 200 inducing points, and the proposed inference with 100 inducing points (Dist GP 100).

the characteristics of the parallel algorithm, which are sometimes overlooked [Gal and Ghahramani, 2014]. One of our stated requirements for a practical parallel inference algorithm is an approximately equal distribution of the load on the nodes. This is especially relevant in a Map-Reduce framework, where the reduce step can only happen after all map computations have finished, so the maximum execution time of one of the workers is the rate limiting step. Figure 3 shows the minimum, maximum and average execution time of all nodes. For 30 cores, there is on average a 1.9% difference between the minimum and maximum run-time of the nodes, suggesting an even distribution of the load.

## 6    GP Regression and Latent Variable Modelling on Real-World Big Data

Next we describe a series of experiments demonstrating the utility in scaling Gaussian processes to big data. We show that GP performance improves with increasing amounts of data in regression and latent variable modelling tasks. We further show that GPs perform better than common models often used for big data.

We evaluate GP regression on the US flight dataset [Hensman et al., 2013] with up to *2 million* points, and compare the results that we got to an array of baselines demonstrating the utility of using GPs for large scale regression. We then present density modelling results over the MNIST dataset, performing imputation tests and digit classification based on model comparison [Titsias and Lawrence, 2010]. As far as we are aware, this is the first GP experiment to run on the full MNIST dataset.

### 6.1    Regression on US Flight Data

In the regression test we predict flight delays from various flight-record characteristics such as flight date and time, flight distance, and others. The US 2008 flight dataset [Hensman et al., 2013] was used with different subset sizes of data: 7K, 70K, and 700K. We selected the first 800K points from the dataset and then split the data randomly into a test set and a training set, using 100K points for testing. We then used the first 7K and 70K points from the large training set to construct the smaller training sets, using the same test set for comparison. This follows the experiment setup of [Hensman et al., 2013] and allows us to compare our results to the Stochastic Variational Inference suggested for GP regression. In addition to that we constructed a 2M points dataset based on a different split using 100K points for test. This test is not comparable to the other experiments due to the non-stationary nature of the data, but it allows us to investigate the performance of the proposed inference compared to the baselines on even larger datasets.

For baselines we predicted the mean of the data, used linear regression, ridge regression with parameter 0.5, and MSE random forest regression at depth 2 with 100 estimators. We report the best results we got for each model for different parameter settings with available resources. We trained our model with 100 inducing points for 500 iterations using LBFGS optimisation and compared the

| Dataset | Mean | Linear | Ridge | RF | Dist GP 100 |
|---|---|---|---|---|---|
| Flight 2M | 38.92 | 37.65 | 37.65 | 37.33 | **35.31** |

Table 2: RMSE for flight data with 2M points by predicting mean, linear regression, ridge regression, random forest regression (RF), and the proposed inference with 100 inducing points (Dist GP).

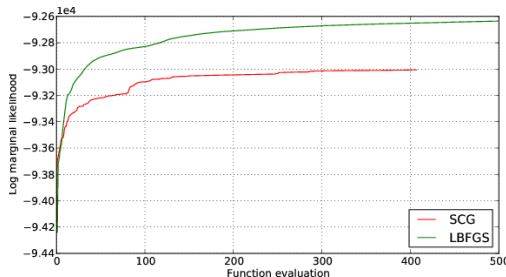

Figure 4: Log likelihood as a function of function evaluation for the 70K flight dataset using SCG and LBFGS optimisation.

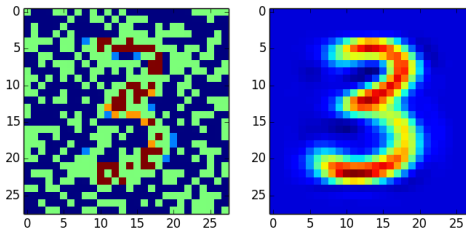

Figure 5: Digit from MNIST with missing data (left) and reconstruction using GPLVM (right).

root mean square error (RMSE) to the baselines as well as SVI with 100 and 200 inducing points (table 1). The results for 2M points are given in table 2. Our inference with 2M data points on a 64 cores machine took $\sim 13.8$ minutes per iteration. Even though the training of the baseline models took several minutes, the use of GPs for big data allows us to take advantage of their desirable properties of uncertainty estimates, robustness to over-fitting, and principled ways for tuning hyper-parameters.

One unexpected result was observed while doing inference with SCG. When increasing the number of data points, the SCG optimiser converged to poor values. When using the final parameters of a model trained on a small dataset to initialise a model to be trained on a larger dataset, performance was as expected. We concluded that SCG was not converging to the correct optimum, whereas L-BFGS performed better (figure 4). We suspect this happens because the modes in the optimisation surface sharpen with more data. This is due to the increased weight of the likelihood terms.

## 6.2 Latent Variable Modelling on MNIST

We also run the GP latent variable model on the full MNIST dataset, which contains 60K examples of 784 dimensions and is considered large in the Gaussian processes community. We trained one model for each digit and used it as a density model, using the predictive probabilities to perform classification. We classify a test point to the model with the highest posterior predictive probability. We follow the calculation in [Titsias and Lawrence, 2010], by taking the ratio of the exponentiated log marginal likelihoods: $p(y^*|Y) = p(y^*, Y)/p(Y) \approx e^{\mathcal{L}_{y^*,Y} - \mathcal{L}_Y}$. Due to the randomness in the initialisation of the inducing inputs and latent point variances, we performed 10 random restarts on each model and chose the model with the largest marginal likelihood lower bound.

We observed that the models converged to a point where they performed similarly, occasionally getting stuck in bad local optima. No pre-processing was performed on the training data as our main aim here is to show the benefit of training GP models using larger amounts of data, rather than proving state-of-the-art performance.

We trained the models on a subset of the data containing 10K points as well as the entire dataset with all 60K points, using additional 10K points for testing. We observed an improvement of 3.03 percentage points in classification error, decreasing the error from **8.98%** to **5.95%**. Training on the full MNIST dataset took 20 minutes for the longest running model, using 500 iterations of SCG. We demonstrate the reconstruction abilties of the GPLVM in figure 5.

## 7 Conclusions

We have scaled sparse GP regression and latent variable modelling, presenting the first distributed inference algorithm able to process datasets with millions of data points. An extensive set of experiments demonstrated the utility in scaling Gaussian processes to big data showing that GP performance improves with increasing amounts of data. We studied the properties of the suggested inference, showing that the inference scales well with data and computational resources, while preserving a balanced distribution of the load among the nodes. Finally, we showed that GPs perform better than many common models used for big data.

The algorithm was implemented in the Map-Reduce architecture and is available as an open-source package, containing an extensively documented implementation of the derivations, with references to the equations presented in the supplementary material for explanation.

## Footnotes

[1]see `http://github.com/markvdw/GParML`

[2]We follow the definition of matrix normal distribution [Arnold, 1981]. For a full treatment of Gaussian Processes, see Rasmussen and Williams [2006].

[3]See Quiñonero-Candela and Rasmussen [2005] for a comprehensive review.

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
