[Supplementary Material]

# Variational Inference in Sparse Gaussian Process Regression and Latent Variable Models – a Gentle Tutorial

Yarin Gal, Mark van der Wilk

September 29, 2014

## 1 Introduction

In this tutorial we will explain the inference procedures developed for the sparse Gaussian process (GP) regression and Gaussian process latent variable model (GPLVM). Due to page limit the derivation given in Titsias (2009) and Titsias & Lawrence (2010) is brief, hence getting a full picture of it requires collecting results from several different sources and a substantial amount of algebra to fill-in the gaps. Our main goal is thus to collect all the results and full derivations into one place to help speed up understanding of this work. In doing so we present a re-parametrisation of the inference that allows it to be carried out in a distributed environment and be scale to huge datasets not commonly handled in the GP community. A secondary goal for this document is, therefore, to accompany our paper and open-source implementation of the parallel inference scheme for the models. We hope that this document will bridge the gap between the equations as implemented in code and those published in the original papers, in order to make it easier to extend existing work. We will assume prior knowledge of Gaussian processes and variational inference, but we also include references for further reading where appropriate.

The paper is organised as follows. In §2 we give a brief review of the sparse GP regression model and the GPLVM. We present the entire derivation of the lower bound of the log marginal likelihood in §3 and §4. In §5 we give some experimental results that extend on the results in the accompanying paper. The derivation of the partial derivatives used in the optimisation is presented in appendix §B with explanations of the techniques used in deriving these. Our implementation of the parallel inference scheme is documented and contains references to the equations in this document for easy adaptation. Since our goal was to create a parallel inference scheme, we do follow a slightly different derivation than that presented in Titsias & Lawrence (2010). We present a re-parametrisation of the models that conditionally decouples the data which allows for the parallel inference. However, the resulting bound is identical to the bound presented in Titsias & Lawrence (2010).

In the appendices we present the derivations of the partial derivatives with respect to the RBF automatic relevance determination (ARD) kernel, and describe the optimisation of the kernel hyper-parameters, locations of the inducing points, and the latent inputs (also referred to as embeddings) – description of which is often overlooked in the literature (this occupies the majority of the paper and intended to be of great help to anyone extending the code for their own use).

## 2 The Gaussian Process Latent Variable Model and Sparse GP Regression – a Quick Review

Here we will quickly review Sparse Gaussian Process Regression and the Gaussian Process Latent Variable Model (GPLVM). We will introduce the structure of the models, the overall procedures required for inference, and the approximations developed to make inference efficient in these models.

## 2.1 Sparse Gaussian Process Regression

### 2.1.1 Gaussian Process Regression

In regression we wish to learn about some function $\mathbf{g}(\mathbf{x})$, given a training dataset consisting of $n$ inputs $\{X_1, \ldots, X_n\}$ and their corresponding outputs $\{F_1, \ldots, F_n\}$. The function is assumed to be $d$ dimensional while the inputs are $q$ dimensional. This data is often written in matrix form for convenience:

$$X \in \mathbb{R}^{n \times q} \tag{2.1}$$

$$F \in \mathbb{R}^{n \times d} \tag{2.2}$$

$$F_i = \mathbf{g}(X_i) \tag{2.3}$$

Here we will adopt the convention that captials denote matrices of data, while subscripted vectors of the same letter will denote a row, i.e. a single data point. For example, $F_i$ denotes the function value of the $i$'th data point, while $F$ denotes the matrix of all given function values. Additionally, functions over matrices will return a matrix of the function evaluated on each row vector.

In regression we often place a Gaussian process prior over the space of functions. This implies a joint Gaussian distribution over all the function values[1], with a covariance matrix known as the "kernel" matrix[2]. For multivariate functions, each dimension will be modelled by a separate GP.

$$\mathbf{g_d} \sim \mathcal{GP}(\mu(\mathbf{x}), k(\mathbf{x}, \mathbf{x}')) \tag{2.4}$$

$$K_{ij} = k(\mathbf{x}_i, \mathbf{x}_j) \tag{2.5}$$

$$p(F|X) = \mathcal{N}(F; \mu(X), K) \tag{2.6}$$

$$= \frac{\exp\left(-\frac{1}{2}\mathrm{Tr}\left[(F - \mu(X))^T K^{-1}(F - \mu(X))\right]\right)}{(2\pi)^{nd/2}|K|^{d/2}} \tag{2.7}$$

It may be the case that we can only obtain noisy evaluations of the function. In this case we introduce a new variable $Y$ containing the noisy observations, making the function values $F$ latent. We assume that the noise on each observation is i.i.d Gaussian, with noise precision $\beta$,

$$p(Y|F) = \frac{\exp\left(-\frac{\beta}{2}\mathrm{Tr}\left[(Y - F)^T(Y - F)\right]\right)}{(2\pi\beta^{-1})^{nd/2}}. \tag{2.8}$$

We will assume reasonable familiarity with GPs and the expressions for their predictive distributions and marginal likelihoods for the rest of the tutorial (Rasmussen & Williams, 2006).

### 2.1.2 Sparse GP Regression

Evaluating $p(Y|X)$ directly[3] is an expensive operation that involves the inversion of the $n$ by $n$ matrix $K$ – thus requiring $\mathcal{O}(n^3)$ time complexity. In order to reduce the computational complexity, Snelson & Ghahramani (2006) suggested the use of a collection of $m$ "inducing points" – a set of points lying in the same input space with corresponding values in the output space. These inducing points aim to summarise the characteristics of the function using less points than the training data. Intuitively, the collection of all training points may contain lots of redundancy, as many points may be given in uninteresting regions.

Consider the following example: the underlying function we are trying to model is a simple linear function with a "kink" near the origin. Only few points are needed to adequately capture the behaviour in flat regions, hence a large number of such points will not improve the posterior much, while still greatly increasing the computational requirements. The use of many points near the kink and only a handful on the flat regions seems more reasonable. By using a finite number of points to describe the function and optimising over their values or locations we can get a more succinct description of the function.

We now define some additional notation that will be used throughout the paper. We let $Z$ denote the locations of the inducing points, an $m$ by $q$ matrix when we have $m$ inducing points. We let $\mathbf{u}$ denote the inferred values of the points, an $m$ by $d$ matrix. We further define $K_{mm}$ to be the covariance matrix

over the $m$ inducing points locations $Z$, and denote by $k_{*m}$ the covariance matrix between point $X^*$ and the points $Z$. Similarly we denote $K_{nm}$ the covariance matrix between the input points $X$ of dimension $n$ and the inducing points of dimension $m$. Prediction now corresponds to taking the GP posterior using only the inducing points instead of the whole training set, which requires only $\mathcal{O}(m^3)$ time complexity.

$$p(F^*|X^*, Y, X) \approx \int \mathcal{N}\left(F^*; k_{*m}K_{mm}^{-1}\mathbf{u}, k_{**} - k_{*m}K_{mm}^{-1}k_{m*}\right) p(\mathbf{u}|Z, Y, X)\mathrm{d}\mathbf{u} \qquad (2.9)$$

where $k(\cdot, \cdot)$ is a covariance function.

Learning the conditional Gaussian distribution over the values of the inducing points requires a simplifying approximation to be made on $p(F|X, \mathbf{u}, Z)$, i.e. how the training data relates to the inducing points. One example is assuming the deterministic relationship $F = K_{nm}K_{mm}^{-1}\mathbf{u}$, giving a computational complexity of $\mathcal{O}(nm^2)$[4]. Quiñonero-Candela & Rasmussen (2005) view this procedure as changing the prior to make inference more tractable, with $Z$ as hyperparameters which can be tuned by maximising the marginal likelihood. On the other hand, Titsias (2009) takes the view of this being a *variational* approximation, with $Z$ as variational parameters. This gives the marginal likelihood above an alternative interpretation as a lower bound on the exact marginal likelihood. Again, the $Z$ values can be optimised over to tighten the lower bound. The detailed derivation will be discussed in §3.

## 2.2 Gaussian Process Latent Variable Models

We can also consider the *unsupervised* equivalent of GPs: the Gaussian Process Latent Variable Model (GPLVM). This model can be used for non-linear dimensionality reduction (Lawrence, 2005). The model setup is the same as the regression case, only that $X$ is unobserved. We assume a prior over the latent $X$ and attempt to infer both the mapping from $X$ to $Y$ and the distribution over $X$ at the same time.

$$X_i \sim \mathcal{N}(X_i; 0, I_q) \qquad (2.10)$$

$$F(X_i) \sim \mathcal{GP}(0, k(X, X)) \qquad (2.11)$$

$$Y_i \sim \mathcal{N}(Y_i; F_i, \beta^{-1}I_d) \qquad (2.12)$$

When the GPLVM model was first introduced it was suggested to optimise over $X$ and perform MAP inference. More recently and relevant to this tutorial, a Variational Bayes approximation was developed by Titsias & Lawrence (2010), using much of the same techniques as for Variational Sparse GPs. In fact, sparse GPs can be seen as a special case of the GPLVM where the inputs are given zero variance.

The main derivation revolves around finding a variational lower bound to:

$$p(Y) = \int p(Y|F)p(F|X)p(X)\mathrm{d}(F, X) \qquad (2.13)$$

$$(2.14)$$

Which then leads to a Gaussian approximation to the posterior $q(X) \approx p(X|Y)$. All this is explained in detail in §4.

# 3 Sparse GPs and the Conditional Independence of the Data

In sparse GPs we use a GP over the inducing points (here denoted $\mathbf{u}$) at some locations (here denoted $Z$) in the input space to approximate the full GP posterior. This can be done by variational approximation where we try to minimise the Kullback–Leibler divergence between the approximating distribution (in our case, the GP over the inducing points – i.e. the locations and values of the inducing points as well as the GP hyper-parameters) and the true distribution we are interested in. This is equivalent to lower bounding the log-marginal likelihood of the true distribution and maximising the lower bound with respect to the variational parameters. In this section we will find the lower bound for the sparse GP using the tools used in (Titsias, 2009). However, with the aim of parallelising computation, we will exploit the conditional independence of the data given the inducing points, and re-parametrise the bound to factorise the marginal likelihood into independent terms.

## 3.1 Introducing the Variational Distribution

We start with the general expression for the log marginal likelihood of the model, after introducing the inducing points, with the distributions factorised using the chain rule:

$$\log p(Y|X) = \log \int p(Y|F)p(F|X,\mathbf{u})p(\mathbf{u})\mathrm{d}(\mathbf{u},F) \tag{3.1}$$

We then introduce a free-form variational distribution $q(\mathbf{u})$ over the inducing points by multiplying the value inside the integral with $\frac{q(\mathbf{u})}{q(\mathbf{u})}$. Using Jensen's inequality (Bishop, 2006, p. 56), we move the log, which is a concave function, into the integral, while keeping $p(F|X,\mathbf{u})q(\mathbf{u})$ outside. After re-arranging the terms to group together all terms containing $F$, we get the following lower bound:

$$\log p(Y|X) \geq \int p(F|X,\mathbf{u})q(\mathbf{u})\log\frac{p(Y|F)p(\mathbf{u})}{q(\mathbf{u})}\mathrm{d}(\mathbf{u},F) \tag{3.2}$$

$$= \int q(\mathbf{u})\left(\int p(F|X,\mathbf{u})\log p(Y|F)\mathrm{d}(F) + \log\frac{p(\mathbf{u})}{q(\mathbf{u})}\right)\mathrm{d}(\mathbf{u}) \tag{3.3}$$

It should be noted that all distributions that involve $\mathbf{u}$ depend on $Z$ as well, which we have omitted in our notation for brevity. In the interpretation used here, $Z$ is not a model parameter, but instead a variational parameter. If $Z$ coincides with $X$, the optimal distribution for $q(\mathbf{u})$ would be the true function posterior $p(F|Y) = \frac{p(Y|F)p(F|X)}{P(Y|X)}$, which would make the bound tight. When $m \leq n$, the inducing point locations will have to be optimised to make the approximation as good as possible.

As a consequence of introducing the variational approximation $q(\mathbf{u}) \approx p(F|Y)$, the function values are *decoupled from each other given the inducing points*. If we decompose $Y$ into the individual data points $(Y_1; Y_2; ...; Y_n)$ with $Y_i \in \mathbb{R}^{1\times d}$ and similarly for $F$, we can write the lower bound as a sum over the data points, since the $Y_i$ are independent of $F_j$ for $j \neq i$:

$$\int p(F|X,\mathbf{u})\log p(Y|F)\mathrm{d}(F) = \int p(F|X,\mathbf{u})\sum_{i=1}^{n}\log p(Y_i|F_i)\mathrm{d}(F) \tag{3.4}$$

$$= \sum_{i=1}^{n}\int p(F_1,..,F_n|X,\mathbf{u})\log p(Y_i|F_i)\mathrm{d}(F_1,...,F_n) \tag{3.5}$$

$$= \sum_{i=1}^{n}\int\log p(Y_i|F_i)\left(\int p(F_1,..,F_n|X,\mathbf{u})\mathrm{d}(F_1,...,F_{i-1},F_{i+1},...,F_n)\right)\mathrm{d}(F_i) \tag{3.6}$$

$$= \sum_{i=1}^{n}\int p(F_i|X,\mathbf{u})\log p(Y_i|F_i)\mathrm{d}(F_i) \tag{3.7}$$

$$= \sum_{i=1}^{n}\int p(F_i|X_1,...,X_n,\mathbf{u})\log p(Y_i|F_i)\mathrm{d}(F_i) \tag{3.8}$$

$$= \sum_{i=1}^{n}\int p(F_i|X_i,\mathbf{u})\log p(Y_i|F_i)\mathrm{d}(F_i) \tag{3.9}$$

where in the transition from line 3 to line 4 we integrate $p(F_1,..,F_n|X,\mathbf{u})$ over $F_1,...,F_{i-1},F_{i+1},...,F_n$ obtaining $p(F_i|X,\mathbf{u})$.

Simplifying each term of the sum by expanding $p(Y_i|F_i)$ as:

$$p(Y_i|F_i) = \mathcal{N}(Y_i; F_i, \beta^{-1}I) = (2\pi\beta^{-1})^{-d/2}\exp(-\frac{\beta}{2}\mathrm{Tr}((Y_i-F_i)^T(Y_i-F_i))) \tag{3.10}$$

$$= (2\pi\beta^{-1})^{-d/2}\exp(-\frac{\beta}{2}(Y_i-F_i)(Y_i-F_i)^T) \tag{3.11}$$

We obtain:

$$\int p(F_i|X_i,\mathbf{u})\log p(Y_i|F_i)\mathrm{d}(F_i) = \int p(F_i|X_i,\mathbf{u})(-\frac{d}{2}\log(2\pi\beta^{-1}) - \frac{\beta}{2}(Y_iY_i^T - 2F_iY_i^T + F_iF_i^T))\mathrm{d}(F_i) \tag{3.12}$$

$$= -\frac{d}{2}\log(2\pi\beta^{-1}) - \frac{\beta}{2}(Y_iY_i^T - 2\langle F_i\rangle_{p(F_i|X_i,\mathbf{u})}Y_i^T + \langle F_iF_i^T\rangle_{p(F_i|X_i,\mathbf{u})})) \tag{3.13}$$

where we use triangular brackets $\langle F\rangle_{q(F)}$ to denote the expectation of $F$ with respect to the distribution $q(F)$.

Now, denoting the covariance matrix between data-point $i$ and the inducing points locations $Z$ as $K_{im}$, we get[5]:

$$\langle F_i\rangle_{p(F_i|X_i,\mathbf{u})} = K_{im}K_{mm}^{-1}\mathbf{u} \tag{3.14}$$

and by noting that $\langle(X - \langle X\rangle)(X - \langle X\rangle)^T\rangle = \langle XX^T - 2\langle X\rangle X^T + \langle X\rangle\langle X\rangle^T\rangle = \langle XX^T\rangle - \langle X\rangle\langle X\rangle^T$,

$$\langle F_iF_i^T\rangle_{p(F_i|X_i,\mathbf{u})} = \langle(F_i - \langle F_i\rangle)(F_i - \langle F_i\rangle)^T\rangle_{p(F_i|X_i,\mathbf{u})} + \langle F_i\rangle_{p(F_i|X_i,\mathbf{u})}\langle F_i\rangle_{p(F_i|X_i,\mathbf{u})}^T \tag{3.15}$$

$$= \left\langle\sum_d(F_{id} - \langle F_{id}\rangle)^2\right\rangle_{p(F_i|X_i,\mathbf{u})} + \langle F_i\rangle_{p(F_i|X_i,\mathbf{u})}\langle F_i\rangle_{p(F_i|X_i,\mathbf{u})}^T \tag{3.16}$$

$$= d\cdot\mathrm{cov}(F_i) + \langle F_i\rangle_{p(F_i|X_i,\mathbf{u})}\langle F_i\rangle_{p(F_i|X_i,\mathbf{u})}^T \tag{3.17}$$

where (Bishop, 2006, p. 87)

$$\mathrm{cov}(F_i) = k(X_i, X_i) - K_{im}K_{mm}^{-1}K_{mi} \tag{3.18}$$

These follow the normal rules of conditional Gaussian distributions explained in detail in Bishop (2006, pp. 85-87).

Therefore, combining equations 3.12, 3.14 and 3.15, we obtain:

$$\int p(F_i|X_i,\mathbf{u})\log p(Y_i|F_i)\mathrm{d}(F_i) = -\frac{d}{2}\log(2\pi\beta^{-1}) - \frac{\beta}{2}\bigg(Y_iY_i^T - 2Y_i\mathbf{u}^TK_{mm}^{-1}K_{mi} \tag{3.19}$$

$$+ K_{im}K_{mm}^{-1}\mathbf{u}\mathbf{u}^TK_{mm}^{-1}K_{mi} + d\cdot k(X_i, X_i) - d\cdot K_{im}K_{mm}^{-1}K_{mi}\bigg) \tag{3.20}$$

## 3.2 Deriving the Optimal Form of $q(\mathbf{u})$

Next, we would like to analytically find optimal $\mathbf{u}$ to use in this equation. Define the lower bound as $\mathcal{F}$:

$$\log p(Y|X) \geq \int q(\mathbf{u})\bigg(\sum_{i=1}^n\int p(F_i|X_i,\mathbf{u})\log p(Y_i|F_i)\mathrm{d}(F_i) + \log\frac{p(\mathbf{u})}{q(\mathbf{u})}\bigg)\mathrm{d}(\mathbf{u}) := \mathcal{F} \tag{3.21}$$

Then, using calculus of variations and Lagrange multipliers (see Bishop (2006, pp. 703-710) for a quick review) we can find the optimal function $q$:

$$\frac{\mathrm{d}(\mathcal{F} + \lambda(\int q(\mathbf{u})\mathrm{d}\mathbf{u} - 1))}{\mathrm{d}q(\mathbf{u})} = \bigg(\sum_{i=1}^n\int p(F_i|X_i,\mathbf{u})\log p(Y_i|F_i)\mathrm{d}(F_i) + \log\frac{p(\mathbf{u})}{q(\mathbf{u})}\bigg) - 1 + \lambda = 0 \tag{3.22}$$

where we differentiated with respect to the integral to get the first term, we obtained the second term $(-1)$ from the derivative with respect to $q$ of the terms inside the integral, and the last term is contributed by the Lagrange multiplier.

Therefore, using properties of the trace operator (see Bishop (2006, p. 696) for example) and by isolating $q$ on the left hand side of the equation, we obtain by plugging eq. 3.19 into the expression above:

$$q(\mathbf{u}) = e^{\lambda-1}e^{\sum_{i=1}^n\int p(F_i|X_i,\mathbf{u})\log p(Y_i|F_i)\mathrm{d}(F_i)}p(\mathbf{u}) \tag{3.23}$$

$$= \exp\left\{ \mathrm{Tr}\left( \mathbf{u}^T \left( -\frac{\beta}{2} \sum_{i=1}^{n} K_{mm}^{-1} K_{mi} K_{im} K_{mm}^{-1} - \frac{1}{2} K_{mm}^{-1} \right) \mathbf{u} + \mathbf{u}^T \left( \beta \sum_{i=1}^{n} K_{mm}^{-1} K_{mi} Y_i \right) + \ldots \right) \right\} \tag{3.24}$$

where we used "..." to denote terms which are used for the normalisation of the distribution but not for the quadratic part, i.e. do not depend on $\mathbf{u}$.

Next, we will need to make use of the following identity:

$$(C + CDC)^{-1} = ((I + CD)C)^{-1} = C^{-1}(I + CD)^{-1} = C^{-1}(C(C^{-1} + D))^{-1} = C^{-1}(C^{-1} + D)^{-1} C^{-1}$$

Since the distribution belongs to the exponential family, we get that it must be a Gaussian distribution with parameters:

$$\Sigma^{-1} = K_{mm}^{-1} + K_{mm}^{-1} \left( \beta \sum_{i=1}^{n} K_{mi} K_{im} \right) K_{mm}^{-1} \tag{3.25}$$

and from from the identity above we get (using $C = K_{mm}^{-1}$ and $D = \beta \sum_{i=1}^{n} K_{mi} K_{im}$)

$$\Sigma = K_{mm} \overbrace{\left( K_{mm} + \beta \sum_{i=1}^{n} K_{mi} K_{im} \right)}^{:=A}{}^{-1} K_{mm} \tag{3.26}$$

and

$$\mu = \beta K_{mm} A^{-1} \underbrace{\sum_{i=1}^{n} K_{mi} Y_i}_{:=B} \tag{3.27}$$

with $q(\mathbf{u}) = \mathcal{N}(\mathbf{u}; \mu, \Sigma)$. Notice the definition of $A$ and $B$ that will be used in the following derivations.

## 3.3  Forming the Evidence Lower Bound

Now, by definition of the prior,

$$\log p(\mathbf{u}) = \log 2\pi^{-nd/2} - \frac{d}{2} \log |K_{mm}| - \frac{1}{2} \mathrm{Tr}(\mathbf{u}^T K_{mm}^{-1} \mathbf{u}) \tag{3.28}$$

To plug this into $\mathcal{F}$, we evaluate $\log q(\mathbf{u})$ as well using equations 3.26 and 3.27

$$\log q(\mathbf{u}) = \mathcal{N}(\mathbf{u}; \mu, \Sigma) = \tag{3.29}$$

$$\log 2\pi^{-nd/2} - \frac{d}{2} \log |K_{mm} A^{-1} K_{mm}| \tag{3.30}$$

$$+ \mathrm{Tr}\left( -\frac{1}{2} \mathbf{u}^T (K_{mm}^{-1} + K_{mm}^{-1} \left( \beta \sum_{i=1}^{n} K_{mi} K_{im} \right) K_{mm}^{-1}) \mathbf{u} \right. \tag{3.31}$$

$$+ \mathbf{u}^T (K_{mm}^{-1} A K_{mm}^{-1})(\beta K_{mm} A^{-1} B) \tag{3.32}$$

$$\left. - \frac{1}{2} \beta^2 (B^T A^{-1} K_{mm})(K_{mm}^{-1} A K_{mm}^{-1})(K_{mm} A^{-1} B) \right) \tag{3.33}$$

$$= \log 2\pi^{-nd/2} - \frac{d}{2} \log |K_{mm} A^{-1} K_{mm}| \tag{3.34}$$

$$- \frac{1}{2} \mathrm{Tr}(\mathbf{u}^T (\beta K_{mm}^{-1} \left( \beta \sum_{i=1}^{n} K_{mi} K_{im} \right) K_{mm}^{-1}) \mathbf{u}) - \frac{1}{2} \mathrm{Tr}(\mathbf{u}^T K_{mm}^{-1} \mathbf{u}) + \beta \mathrm{Tr}(\mathbf{u}^T K_{mm}^{-1} B) \tag{3.35}$$

$$- \frac{1}{2} \beta^2 \mathrm{Tr}(B^T A^{-1} B) \tag{3.36}$$

where we used the symmetry of the covariance matrices in the second transition.

Finally, by collecting equations 3.29, 3.28, and 3.19 into 3.21, **u** eliminates (following trace properties again), and we obtain

$$\log p(Y|X) \geq \frac{d}{2} \log |K_{mm}| - \frac{d}{2} \log \left| K_{mm} + \beta \sum_{i=1}^{n} K_{mi} K_{im} \right| - \frac{nd}{2} \log 2\pi \beta^{-1} \tag{3.37}$$

$$- \frac{\beta}{2} \sum_{i=1}^{n} \left( Y_i Y_i^T + d \cdot k(X_i, X_i) - d \cdot \mathrm{Tr}(K_{mm}^{-1} K_{mi} K_{im}) \right) \tag{3.38}$$

$$+ \frac{\beta^2}{2} \mathrm{Tr} \left( \left( \sum_{i=1}^{n} K_{mi} Y_i \right)^T \left( K_{mm} + \beta \sum_{i=1}^{n} K_{mi} K_{im} \right)^{-1} \left( \sum_{i=1}^{n} K_{mi} Y_i \right) \right) \tag{3.39}$$

Factorising the log marginal likelihood over the data points. Notice that this is exactly the same bound derived in Titsias (2009), only *factored over the input data.*

## 3.4 Parallel Inference in Sparse GPs

A nice application of the re-parametrisation brought above is the distribution of the inference into independent nodes. In a parallel implementation of the inference, each node $i$ in the parallel implementation has to calculate $Y_i^T K_{im}$, $K_{mi} K_{im}$, $k(X_i, X_i)$, and $Y_i Y_i^T$. These take $\mathcal{O}(m^2 + d^2 + md)$ time complexity, since all the operations involved in the collection of the partial sums are matrix products of matrices of dimensions $m$ by 1 with 1 by $d$ and 1 by $m$, and $d$ by 1 with 1 by $d$, as well as $m$ by $m$ with $m$ by 1 matrix products. Then, we accumulate the results (asynchronously), and perform once $\mathcal{O}(m^3)$ calculations to evaluate the log marginal likelihood. This is repeated at each step of the optimisation over the kernel hyper-parameters and the locations of the inducing points. To optimise the hyper-parameters, we need to differentiate the log evidence with respect to the kernel hyper-parameters ($\sigma$ and $\alpha$ for an RBF kernel), the observation noise ($\beta$), and the locations of the inducing points ($Z$).

We send to all nodes the global parameters $Z$, **k** (kernel hyper-parameters), and $\beta$ for them to calculate the partial terms $Y_i^T K_{im}$, $K_{mi} K_{im}$, $k(X_i, X_i)$, and $Y_i Y_i^T$ and return to the master node ($m \times m \times q$ matrices – constant space complexity for fixed $m$). The master node sums the log evidence using the sum of the partial terms, and then performs optimisation over the global parameters $Z$, **k** and $\beta$. So, we have one MapReduce step[6] transferring information between the master node and slave nodes to follow:

1. The master sends $Z$, **k** and $\beta$ to the nodes

2. The nodes calculate partial $\mathcal{F}$, $\partial \mathcal{F}$ ($m \times m \times q$ matrices) and return to the master

3. The master optimises $Z$, **k**, and $\beta$.

# 4 GPLVMs and the Conditional Independence of the Data

Using the factorisation developed in the previous section we can easily derive a similar lower bound for the GPLVM. This is done by integrating the factorised sparse GP over $X$ and introducing an additional free-form variational distribution $q(X)$ over $X$ – which maintains the mean $\mu_i$ and covariance $S_i$ for each latent variable $X_i$.

Indeed, following the same initial development we can integrate the evidence over $X$ and use Jensen's inequality to get

$$\log p(Y) = \log \int q(X) \frac{p(Y|X)p(X)}{q(X)} \mathrm{d}(X) \tag{4.1}$$

$$\geq \int q(X) \left( \log p(Y|X) + \log \frac{p(X)}{q(X)} \right) \mathrm{d}(X) \tag{4.2}$$

Using eq. 3.21 we can bound $\log p(Y|X)$ from below to get

$$\geq \int q(X) \left( \int q(\mathbf{u}) \left( \sum_{i=1}^{n} \int p(F_i|X_i, \mathbf{u}) \log p(Y_i|F_i) \mathrm{d}(F_i) + \log \frac{p(\mathbf{u})}{q(\mathbf{u})} \right) \mathrm{d}(\mathbf{u}) + \log \frac{p(X)}{q(X)} \right) \mathrm{d}(X) \tag{4.3}$$

and after re-arranging the terms,

$$= \int q(\mathbf{u}) \left( \int q(X) \left( \sum_{i=1}^{n} \int p(F_i|X_i, \mathbf{u}) \log p(Y_i|F_i) \mathrm{d}(F_i) + \sum_{i=1}^{n} \log \frac{p(X_i)}{q(X_i)} \right) \mathrm{d}(X) + \log \frac{p(\mathbf{u})}{q(\mathbf{u})} \right) \mathrm{d}(\mathbf{u}) \quad (4.4)$$

$$= \int q(\mathbf{u}) \left( \int q(X) \left( \sum_{i=1}^{n} \int p(F_i|X_i, \mathbf{u}) \log p(Y_i|F_i) \mathrm{d}(F_i) \right) \mathrm{d}(X) + \log \frac{p(\mathbf{u})}{q(\mathbf{u})} \right) \mathrm{d}(\mathbf{u}) \quad (4.5)$$

$$- \sum_{i=1}^{n} \int q(X_i) \log \frac{q(X_i)}{p(X_i)} \mathrm{d}(X_i) \quad (4.6)$$

where $\int q(X_i) \log \frac{q(X_i)}{p(X_i)} \mathrm{d}(X_i)$ is just the KL-divergence between $q$ and $p$:

$$= \int q(\mathbf{u}) \left( \sum_{i=1}^{n} \int q(X) \left( \int p(F_i|X_i, \mathbf{u}) \log p(Y_i|F_i) \mathrm{d}(F_i) \right) \mathrm{d}(X) + \log \frac{p(\mathbf{u})}{q(\mathbf{u})} \right) \mathrm{d}(\mathbf{u}) - \sum_{i=1}^{n} KL(q(X_i)||p(X_i)) \quad (4.7)$$

which evaluates to (marginalising over $X_j$ for $j \neq i$)

$$= \int q(\mathbf{u}) \left( \sum_{i=1}^{n} \int q(X_i) \left( \int p(F_i|X_i, \mathbf{u}) \log p(Y_i|F_i) \mathrm{d}(F_i) \right) \mathrm{d}(X_i) + \log \frac{p(\mathbf{u})}{q(\mathbf{u})} \right) \mathrm{d}(\mathbf{u}) - \sum_{i=1}^{n} KL(q(X_i)||p(X_i)) \quad (4.8)$$

$$:= \mathcal{F} \quad (4.9)$$

Following the derivations of the previous section, we find optimal $q(\mathbf{u})$ with respect to

$$\sum_{i=1}^{n} \int q(X_i) p(F_i|X_i, \mathbf{u}) \log p(Y_i|F_i) \mathrm{d}(X_i, F_i) \quad (4.10)$$

$$= \sum_{i=1}^{n} \int q(X_i) p(F_i|X_i, \mathbf{u}) (-\frac{d}{2} \log(2\pi\beta^{-1}) - \frac{\beta}{2}(Y_i Y_i^T - 2F_i Y_i^T + F_i F_i^T)) \mathrm{d}(X_i, F_i) \quad (4.11)$$

$$\quad (4.12)$$

$$= \sum_{i=1}^{n} \left( -\frac{d}{2} \log(2\pi\beta^{-1}) - \frac{\beta}{2}(Y_i Y_i^T - 2 \langle F_i \rangle_{p(F_i|X_i, \mathbf{u})q(X_i)} Y_i^T + \langle F_i F_i^T \rangle_{p(F_i|X_i, \mathbf{u})q(X_i)}) \right) \quad (4.13)$$

Where now the optimal distribution is defined in terms of the expectation of the covariance matrices with respect to $X$,

$$\langle F_i \rangle_{p(F_i|X_i, \mathbf{u})q(X_i)} = \left\langle K_{im}^{X_i} \right\rangle_{q(X_i)} K_{mm}^{-1} \mathbf{u} \quad (4.14)$$

and

$$\left\langle F_i F_i^T \right\rangle_{p(F_i|X_i, \mathbf{u})q(X_i)} = d \cdot \mathrm{cov}(F_i) + \langle F_i \rangle_{p(F_i|X_i, \mathbf{u})q(X_i)} \langle F_i \rangle_{p(F_i|X_i, \mathbf{u})q(X_i)}^T \quad (4.15)$$

where

$$\mathrm{cov}(F_i) = \left\langle K_{ii}^{X_i} \right\rangle_{q(X_i)} - \left\langle K_{im}^{X_i} \right\rangle_{q(X_i)} K_{mm}^{-1} \left\langle K_{mi}^{X_i} \right\rangle_{q(X_i)}. \quad (4.16)$$

This optimal $q(\mathbf{u})$ is given by

$$\Sigma = K_{mm} \overbrace{\left( K_{mm} + \beta \sum_{i=1}^{n} \left\langle K_{mi}^{X_i} K_{im}^{X_i} \right\rangle_{q(X_i)} \right)}^{:=A}{}^{-1} K_{mm} \quad (4.17)$$

and

$$\mu = \beta K_{mm} A^{-1} \underbrace{\sum_{i=1}^{n} \left\langle K_{mi}^{X_i} \right\rangle_{q(X_i)} Y_i}_{:=B} \tag{4.18}$$

with $q(\mathbf{u}) = \mathcal{N}(\mathbf{u}; \mu, \Sigma)$.

This evaluates to the following lower bound on the log marginal likelihood

$$\log p(Y) \geq \frac{d}{2} \log |K_{mm}| - \frac{d}{2} \log \left| K_{mm} + \beta \sum_{i=1}^{n} \left\langle K_{mi}^{X_i} K_{im}^{X_i} \right\rangle_{q(X_i)} \right| - \frac{nd}{2} \log 2\pi \beta^{-1} \tag{4.19}$$

$$- \frac{\beta}{2} \sum_{i=1}^{n} \left( Y_i Y_i^T + d \left\langle K_{ii}^{X_i} \right\rangle_{q(X_i)} - d\mathrm{Tr}\left( K_{mm}^{-1} \left\langle K_{mi}^{X_i} K_{im}^{X_i} \right\rangle_{q(X_i)} \right) \right) \tag{4.20}$$

$$+ \frac{\beta^2}{2} \mathrm{Tr}\left( \left( \sum_{i=1}^{n} \left\langle K_{mi}^{X_i} \right\rangle_{q(X_i)} Y_i \right)^T \left( K_{mm} + \beta \sum_{i=1}^{n} \left\langle K_{mi}^{X_i} K_{im}^{X_i} \right\rangle_{q(X_i)} \right)^{-1} \cdot \left( \sum_{i=1}^{n} \left\langle K_{mi}^{X_i} \right\rangle_{q(X_i)} Y_i \right) \right) \tag{4.21}$$

$$- \sum_{i=1}^{n} KL(q(X_i)||p(X_i)) \tag{4.22}$$

## 4.1   Parallel Inference in GPLVMs

Similarly, a parallel inference algorithm can be derived based on this factorisation. First, we send to all nodes the global parameters $Z$, $\mathbf{k}$ (the kernel hyper-parameters), and $\beta$ for them to calculate the partial terms $\left\langle K_{mi}^{X_i} \right\rangle_{q(X_i)} Y_i$, $\left\langle K_{mi}^{X_i} K_{im}^{X_i} \right\rangle_{q(X_i)}$, $\left\langle K_{ii}^{X_i} \right\rangle_{q(X_i)}$, and $Y_i Y_i^T$ and return to the master node ($m \times m \times q$ matrices – constant for fixed $m$). The master node then sends the accumulated partial terms back to the nodes and performs global optimisation over $Z$ and $\mathbf{k}$ and $\beta$. At the same time the nodes perform local optimisation on $\mu$ and $S$, the embedding posterior parameters, which can be carried out by parallelising scaled conjugate gradient (SCG) or using local gradient descent. So, we have two "MapReduce" steps for the master node $\leftrightarrow$ slave nodes to follow:

1. $Z$, $\mathbf{k}$, $\beta$ $\rightarrow$

2. $\leftarrow$ partial $\mathcal{F}$, $\partial\mathcal{F}$ ($m \times m \times q$ matrices)

3. whole $\mathcal{F}$, $\partial\mathcal{F}$ $\rightarrow$

4. optimise $Z$, $\mathbf{k}$, $\beta$ $\leftrightarrow$ optimise $\mu_i$ and $S_i$

In the appendices we cover the derivations of the partial derivatives with respect to the global variables as well as the local ones.

# 5   Discussion and Experimental Results

The derivation of the partial derivatives is only the first step in the implementation of the inference. The actual development of new models based on the models presented above requires some know-how (such as embeddings initialisation) and other intricates. This will be discussed here in future revisions of the tutorial.

Next we give some experimental results that extend on the results in the accompanying paper.

## 5.1   Comparison to GPy

We compare our latent space to GPy, which we use as a reference implementation. Just like in the original paper (Titsias & Lawrence, 2010), we used the oil-flow dataset. Both algorithms were run until no significant improvement in the marginal likelihood was found.

The two latent spaces are shown in figure 1. The latent spaces are qualitatively similar, but differ due to a slightly different implementation of the optimiser. Like the results in Titsias & Lawrence (2010) all but one of the ARD parameters decrease to zero, giving an effectively 1D latent space.

Figure 1: Latent space produced by the parallel inference (left) and GPy (right) using the oilflow dataset (Titsias & Lawrence, 2010).

## 5.2 Robustness to Node Failure

One desirable characteristic of a parallel inference scheme is robustness to failure of nodes. One way of dealing with this would be to load the data to a different node and restart the calculation. However, since the speed of one iteration is limited by the *slowest* calculation on one of the nodes, this could slow down the algorithm by the time it takes to load the intermediate data onto the new node. An alternative strategy would be to drop the partial term from the calculation and use a slightly noisy gradient calculation in the optimisation for one iteration. Here we investigate the robustness of our inference to this procedure.

Figure 2: **Node failure test,** for node failure frequencies of 0%, 1% and 2% per *iteration*. Shown is the average log marginal likelihood as a function of the iteration for 500 iterations.

We ran our parallel inference on the oil-flow dataset using the same setting as above for 500 iterations accumulating the log marginal likelihood as a function of the iteration. We used 10 nodes and simulated failure frequencies of 0%, 1% and 2% per *iteration*. The experiment was repeated 10 times and the log marginal likelihood averaged. Even a failure rate of 1% per iteration for 500 iterations translates to a high number of 1 out the 10 nodes failing on average every 10 iterations.

As we observe in figure 2 a node failure frequency of 1% hurts total performance by decreasing the log marginal likelihood from -1500 to -5000 on average. It seems that a higher failure frequency leads to convergence to worse local optima or a failure of the optimiser, possibly because of the finite

differences approximation to the function curvature used by SCG, which might suffer from noisy gradient estimations. It is also interesting to note that the embeddings discovered are less pronounced than the ones shown in figure 1 but still have only one major latent dimension. For 0% failure rate the ARD parameters are 0.02 for all but one dimension (0.15), for 1% failure rate the ARD parameters are 0.10 for all but one dimension (0.17), and for 2% failure rate the ARD parameters are 0.29 for all but one dimension (0.34).

# A  Selection of kernel

For all tasks in this paper we use the RBF Automatic Relevance Determination (ARD) kernel. This kernel is given by the following formula:

$$k(x, x') = \sigma_f^2 \exp\left(-\frac{1}{2}\sum_{q=1}^{Q} \alpha_q (x_q - x'_q)^2\right). \tag{A.1}$$

For this choice of kernel, we can evaluate the analytic solutions to the statistics $\left\langle K_{ii}^{X_i}\right\rangle_{q(X_i)}$, $\left\langle K_{mi}^{X_i}\right\rangle_{q(X_i)}$, and $\left\langle K_{mi}^{X_i} K_{im}^{X_i}\right\rangle_{q(X_i)}$ easily.

First of all, we have

$$\left\langle K_{ii}^{X_i}\right\rangle_{q(X_i)} = \sigma_f^2. \tag{A.2}$$

Next, for $\left\langle K_{mi}^{X_i}\right\rangle_{q(X_i)}$ we have

$$\left(\left\langle K_{mi}^{X_i}\right\rangle_{q(X_i)}\right)_j = \int \sigma_f^2 \exp\left(-\frac{1}{2}(X_i - Z_j)^T \alpha (X_i - Z_j)\right) \frac{1}{|S|^{1/2}(2\pi)^{Q/2}} \tag{A.3}$$

$$\cdot \exp\left\{-\frac{1}{2}\left((X_i - \mu_i)^T S_i^{-1}(X_i - \mu_i)\right)\right\}\mathrm{d}(X_i) \tag{A.4}$$

$$= \frac{\sigma_f^2}{|S|^{1/2}(2\pi)^{Q/2}} \tag{A.5}$$

$$\cdot \int \exp\left\{-\frac{1}{2}\left(X_i^T \alpha X_i - 2X_i^T \alpha Z_j + Z_j^T \alpha Z_j + X_i^T S_i^{-1} X_i - 2X_i^T S_i^{-1}\mu_i + \mu_i^T S_i^{-1}\mu_i\right)\right\}\mathrm{d}(X_i) \tag{A.6}$$

$$= \frac{\sigma_f^2}{|S|^{1/2}(2\pi)^{Q/2}} \tag{A.7}$$

$$\cdot \int \exp\left\{-\frac{1}{2}\left(X_i^T (\alpha + S_i^{-1}) X_i - 2X_i^T (\alpha Z_j + S_i^{-1}\mu_i) + Z_j^T \alpha Z_j + \mu_i^T S_i^{-1}\mu_i\right)\right\}\mathrm{d}(X_i) \tag{A.8}$$

$$= \frac{\sigma_f^2}{|S|^{1/2}|\alpha + S_i^{-1}|^{1/2}\cdot(2\pi)^{Q/2}|\alpha + S_i^{-1}|^{-1/2}} \int \exp\left\{-\frac{1}{2}\Bigg[ \right. \tag{A.9}$$

$$\left(X_i - (\alpha + S_i^{-1})^{-1}(\alpha Z_j + S_i^{-1}\mu_i)\right)^T \left(\alpha + S_i^{-1}\right)\left(X_i - (\alpha + S_i^{-1})^{-1}(\alpha Z_j + S_i^{-1}\mu_i)\right) \tag{A.10}$$

$$\left. - (\alpha Z_j + S_i^{-1}\mu_i)^T(\alpha + S_i^{-1})^{-1}(\alpha Z_j + S_i^{-1}\mu_i) + Z_j^T \alpha Z_j + \mu_i^T S_i^{-1}\mu_i\Bigg]\right\}\mathrm{d}(X_i) \tag{A.11}$$

$$= \frac{\sigma_f^2}{|S_i|^{1/2}|\alpha + S_i^{-1}|^{1/2}} \exp\left\{-\frac{1}{2}\sum_{q=1}^{Q} \frac{1}{S_{iq}\alpha_q + 1}\left(-S_{iq}\alpha_q^2 Z_{jq}^2 - 2\alpha_q \mu_{iq} Z_{jq} - S_{iq}^{-1}\mu_{iq}^2 \right. \right. \tag{A.12}$$

$$\left. \left. + Z_{jq}^2 \alpha_q + Z_{jq}^2 \alpha_q^2 S_{iq} + \mu_{iq}^2 S_{iq}^{-1} + \mu_{iq}^2 \alpha_q\right)\right\} \tag{A.13}$$

$$= \frac{\sigma_f^2}{\prod_{q=1}^{Q}(S_{iq}\alpha_q + 1)^{1/2}} \exp\left\{-\frac{1}{2}\sum_{q=1}^{Q} \frac{(Z_{jq} - \mu_{iq})^2 \alpha_q}{S_{iq}\alpha_q + 1}\right\} \tag{A.14}$$

Lastly, for $\left\langle K_{mi}^{X_i} K_{im}^{X_i} \right\rangle_{q(X_i)}$ we can derive a similar result

$$\left\langle K_{mi}^{X_i} K_{im}^{X_i} \right\rangle_{q(X_i)} = \sigma_f^4 \prod_{q=1}^{Q} \frac{\exp\left(-\frac{\alpha_q(Z_{mq}-Z_{m'q})^2}{4} - \frac{\alpha_q(\mu_{iq}-\frac{Z_{mq}}{2}-\frac{Z_{m'q}}{2})^2}{2\alpha_q S_{iq}+1}\right)}{(2\alpha_q S_{iq}+1)^{1/2}} \tag{A.15}$$

# B  Optimising the kernel hyper-parameters and locations of the inducing points

In order to perform inference in the variational setting, we need to perform optimisation over the log likelihood lower bound with respect to the kernel hyper-parameters and the locations of the inducing points. In addition to that, for the latent variable model we also need to perform optimisation over the means and covariances of the latent $X$ points. For that, we need to obtain the partial derivatives of the lower bound on the log marginal likelihood $\mathcal{F}$ with respect to each of the variables to be optimised.

$$\mathcal{F} = -\frac{nd}{2}\log 2\pi + \frac{dn}{2}\log\beta + \frac{d}{2}\log|K_{mm}| - \frac{d}{2}\log\left|K_{mm} + \beta\sum_{i=1}^{n}\left\langle K_{mi}^{X_i} K_{im}^{X_i}\right\rangle_{q(X_i)}\right| \tag{B.1}$$

$$-\frac{\beta}{2}\sum_{i=1}^{n} Y_i Y_i^T - \frac{\beta d}{2}\sum_{i=1}^{n}\left\langle K_{ii}^{X_i}\right\rangle_{q(X_i)} + \frac{\beta d}{2}\mathrm{Tr}\left(K_{mm}^{-1}\sum_{i=1}^{n}\left\langle K_{mi}^{X_i} K_{im}^{X_i}\right\rangle_{q(X_i)}\right) \tag{B.2}$$

$$+\frac{\beta^2}{2}\mathrm{Tr}\left(\left(\sum_{i=1}^{n}\left\langle K_{mi}^{X_i}\right\rangle_{q(X_i)} Y_i\right)^T \left(K_{mm} + \beta\sum_{i=1}^{n}\left\langle K_{mi}^{X_i} K_{im}^{X_i}\right\rangle_{q(X_i)}\right)^{-1}\right. \tag{B.3}$$

$$\left. \cdot \left(\sum_{i=1}^{n}\left\langle K_{mi}^{X_i}\right\rangle_{q(X_i)} Y_i\right)\right) \tag{B.4}$$

$$-\sum_{i=1}^{n} KL(q(X_i)\|p(X_i)) \tag{B.5}$$

$$\tag{B.6}$$

We thus need to optimise over the variables $Z$, $(\mu_i, S_i)_{i\leq N}$, $\beta$, and $\theta = (\sigma_f^2, \alpha_1, ..., \alpha_Q)$. Following the chain rule for multivariate functions [wiki: chain rule], we get that the derivative of the lower bound of the log marginal likelihood $\mathcal{F}$ for $Z_{jk}$, for example, is given by:

$$\frac{\partial\mathcal{F}}{\partial Z_{jk}} = \frac{\partial\mathcal{F}}{\partial K_{mm}}\frac{\partial K_{mm}}{\partial Z_{jk}} \tag{B.7}$$

$$+ \frac{\partial\mathcal{F}}{\partial\left(\sum_{i=1}^{n}\left\langle K_{ii}^{X_i}\right\rangle_{q(X_i)}\right)}\frac{\partial\left(\sum_{i=1}^{n}\left\langle K_{ii}^{X_i}\right\rangle_{q(X_i)}\right)}{\partial Z_{jk}} \tag{B.8}$$

$$+ \frac{\partial\mathcal{F}}{\partial\left(\sum_{i=1}^{n}\left\langle K_{mi}^{X_i}\right\rangle_{q(X_i)} Y_i\right)}\frac{\partial\left(\sum_{i=1}^{n}\left\langle K_{mi}^{X_i}\right\rangle_{q(X_i)} Y_i\right)}{\partial Z_{jk}} \tag{B.9}$$

$$+ \frac{\partial\mathcal{F}}{\partial\left(\sum_{i=1}^{n}\left\langle K_{mi}^{X_i} K_{im}^{X_i}\right\rangle_{q(X_i)}\right)}\frac{\partial\left(\sum_{i=1}^{n}\left\langle K_{mi}^{X_i} K_{im}^{X_i}\right\rangle_{q(X_i)}\right)}{\partial Z_{jk}} \tag{B.10}$$

where when differentiating with respect to $(\mu_i, S_i)_{i\leq N}$ we also need to find the partial derivative of

$$\sum_{i=1}^{n} KL(q(X_i)\|p(X_i)). \tag{B.11}$$

We have

$$\frac{\partial\left(\sum_{i=1}^{n}\left\langle K_{ii}^{X_i}\right\rangle_{q(X_i)}\right)}{\partial Z_{jk}} = \sum_{i=1}^{n}\left(\frac{\partial\left\langle K_{ii}^{X_i}\right\rangle_{q(X_i)}}{\partial Z_{jk}}\right) \tag{B.12}$$

$$\frac{\partial\left(\sum_{i=1}^{n}\left\langle K_{mi}^{X_i}\right\rangle_{q(X_i)} Y_i\right)}{\partial Z_{jk}} = \sum_{i=1}^{n}\left(\frac{\partial\left\langle K_{mi}^{X_i}\right\rangle_{q(X_i)} Y_i}{\partial Z_{jk}}\right) \tag{B.13}$$

$$\frac{\partial\left(\sum_{i=1}^{n}\left\langle K_{mi}^{X_i} K_{im}^{X_i}\right\rangle_{q(X_i)}\right)}{\partial Z_{jk}} = \sum_{i=1}^{n}\left(\frac{\partial\left\langle K_{mi}^{X_i} K_{im}^{X_i}\right\rangle_{q(X_i)}}{\partial Z_{jk}}\right) \tag{B.14}$$

Thus we only need to look at the partial derivatives inside the sums.

## B.1  Partial derivatives of $\mathcal{F}$

**The partial derivative** $\dfrac{\partial\mathcal{F}}{\partial\left(\sum_{i=1}^{n}\left\langle K_{ii}^{X_i}\right\rangle_{q(X_i)}\right)}$

$$\frac{\partial\mathcal{F}}{\partial\left(\sum_{i=1}^{n}\left\langle K_{ii}^{X_i}\right\rangle_{q(X_i)}\right)} = -\frac{\beta d}{2} \tag{B.15}$$

**The partial derivative** $\dfrac{\partial\mathcal{F}}{\partial\left(\sum_{i=1}^{n}\left\langle K_{mi}^{X_i}\right\rangle_{q(X_i)} Y_i\right)}$   Using properties of the trace operator [wiki: trace]

$$\mathrm{d}\mathcal{F}(\sum_{i=1}^{n}\left\langle K_{mi}^{X_i}\right\rangle_{q(X_i)} Y_i) \tag{B.16}$$

$$= \mathrm{d}\mathrm{Tr}(\frac{\beta^2}{2}\left(\sum_{i=1}^{n}\left\langle K_{mi}^{X_i}\right\rangle_{q(X_i)} Y_i\right)^T (K_{mm} + \beta\left(\sum_{i=1}^{n}\left\langle K_{mi}^{X_i} K_{im}^{X_i}\right\rangle_{q(X_i)}\right))^{-1}\left(\sum_{i=1}^{n}\left\langle K_{mi}^{X_i}\right\rangle_{q(X_i)} Y_i\right)) \tag{B.17}$$

$$= \mathrm{Tr}(\frac{\beta^2}{2}\left(\sum_{i=1}^{n}\left\langle K_{mi}^{X_i}\right\rangle_{q(X_i)} Y_i\right)^T (K_{mm} + \beta\left(\sum_{i=1}^{n}\left\langle K_{mi}^{X_i} K_{im}^{X_i}\right\rangle_{q(X_i)}\right))^{-1}\mathrm{d}\left(\sum_{i=1}^{n}\left\langle K_{mi}^{X_i}\right\rangle_{q(X_i)} Y_i\right)) \tag{B.18}$$

$$+ \mathrm{Tr}(\frac{\beta^2}{2}\mathrm{d}\left(\sum_{i=1}^{n}\left\langle K_{mi}^{X_i}\right\rangle_{q(X_i)} Y_i\right)^T (K_{mm} + \beta\left(\sum_{i=1}^{n}\left\langle K_{mi}^{X_i} K_{im}^{X_i}\right\rangle_{q(X_i)}\right))^{-1}\left(\sum_{i=1}^{n}\left\langle K_{mi}^{X_i}\right\rangle_{q(X_i)} Y_i\right)) \tag{B.19}$$

$$= \mathrm{Tr}(\frac{\beta^2}{2}\left(\sum_{i=1}^{n}\left\langle K_{mi}^{X_i}\right\rangle_{q(X_i)} Y_i\right)^T (K_{mm} + \beta\left(\sum_{i=1}^{n}\left\langle K_{mi}^{X_i} K_{im}^{X_i}\right\rangle_{q(X_i)}\right))^{-1}\mathrm{d}\left(\sum_{i=1}^{n}\left\langle K_{mi}^{X_i}\right\rangle_{q(X_i)} Y_i\right)) \tag{B.20}$$

$$+ \mathrm{Tr}(\frac{\beta^2}{2}(K_{mm} + \beta\left(\sum_{i=1}^{n}\left\langle K_{mi}^{X_i} K_{im}^{X_i}\right\rangle_{q(X_i)}\right))^{-1}\left(\sum_{i=1}^{n}\left\langle K_{mi}^{X_i}\right\rangle_{q(X_i)} Y_i\right)\mathrm{d}\left(\sum_{i=1}^{n}\left\langle K_{mi}^{X_i}\right\rangle_{q(X_i)} Y_i\right)^T) \tag{B.21}$$

$$\tag{B.22}$$

Since

$$\mathrm{Tr}(\frac{\beta^2}{2}\left(\sum_{i=1}^{n}\left\langle K_{mi}^{X_i}\right\rangle_{q(X_i)} Y_i\right)^T (K_{mm} + \beta\left(\sum_{i=1}^{n}\left\langle K_{mi}^{X_i} K_{im}^{X_i}\right\rangle_{q(X_i)}\right))^{-1}\mathrm{d}\left(\sum_{i=1}^{n}\left\langle K_{mi}^{X_i}\right\rangle_{q(X_i)} Y_i\right)) \tag{B.23}$$

$$= \mathrm{Tr}(\frac{\beta^2}{2}\mathrm{d}\left(\sum_{i=1}^{n}\left\langle K_{mi}^{X_i}\right\rangle_{q(X_i)} Y_i\right)^T (K_{mm} + \beta\left(\sum_{i=1}^{n}\left\langle K_{mi}^{X_i} K_{im}^{X_i}\right\rangle_{q(X_i)}\right))^{-1}\left(\sum_{i=1}^{n}\left\langle K_{mi}^{X_i}\right\rangle_{q(X_i)} Y_i\right)) \tag{B.24}$$

$$= \text{Tr}\left(\frac{\beta^2}{2}\left(K_{mm} + \beta\left(\sum_{i=1}^{n}\left\langle K_{mi}^{X_i}K_{im}^{X_i}\right\rangle_{q(X_i)}\right)\right)^{-1}\left(\sum_{i=1}^{n}\left\langle K_{mi}^{X_i}\right\rangle_{q(X_i)}Y_i\right)\text{d}\left(\sum_{i=1}^{n}\left\langle K_{mi}^{X_i}\right\rangle_{q(X_i)}Y_i\right)^T\right)$$

$$\text{(B.25)}$$

We get that the differential equals

$$= \text{Tr}\left(\beta^2\left(K_{mm} + \beta\left(\sum_{i=1}^{n}\left\langle K_{mi}^{X_i}K_{im}^{X_i}\right\rangle_{q(X_i)}\right)\right)^{-1}\left(\sum_{i=1}^{n}\left\langle K_{mi}^{X_i}\right\rangle_{q(X_i)}Y_i\right)\text{d}\left(\sum_{i=1}^{n}\left\langle K_{mi}^{X_i}\right\rangle_{q(X_i)}Y_i\right)^T\right) \quad \text{(B.26)}$$

Therefore,

$$\frac{\partial\mathcal{F}}{\partial\left(\sum_{i=1}^{n}\left\langle K_{mi}^{X_i}\right\rangle_{q(X_i)}Y_i\right)} = \beta^2\left(K_{mm} + \beta\sum_{i=1}^{n}\left\langle K_{mi}^{X_i}K_{im}^{X_i}\right\rangle_{q(X_i)}\right)^{-1}\left(\sum_{i=1}^{n}\left\langle K_{mi}^{X_i}\right\rangle_{q(X_i)}Y_i\right) \quad \text{(B.27)}$$

**The partial derivative** $\dfrac{\partial\mathcal{F}}{\partial\left(\sum_{i=1}^{n}\left\langle K_{mi}^{X_i}K_{im}^{X_i}\right\rangle_{q(X_i)}\right)}$

$$\text{d}\mathcal{F}\left(\left(\sum_{i=1}^{n}\left\langle K_{mi}^{X_i}K_{im}^{X_i}\right\rangle_{q(X_i)}\right)\right) \quad \text{(B.28)}$$

$$= \text{d}\left(-\frac{d}{2}\log Det\left(K_{mm} + \beta\left(\sum_{i=1}^{n}\left\langle K_{mi}^{X_i}K_{im}^{X_i}\right\rangle_{q(X_i)}\right)\right) + \frac{\beta d}{2}\text{Tr}\left(K_{mm}^{-1}\left(\sum_{i=1}^{n}\left\langle K_{mi}^{X_i}K_{im}^{X_i}\right\rangle_{q(X_i)}\right)\right) \quad \text{(B.29)}$$

$$+ \frac{\beta^2}{2}\text{Tr}\left(\left(\sum_{i=1}^{n}\left\langle K_{mi}^{X_i}\right\rangle_{q(X_i)}Y_i\right)\left(\sum_{i=1}^{n}\left\langle K_{mi}^{X_i}\right\rangle_{q(X_i)}Y_i\right)^T\left(K_{mm} + \beta\left(\sum_{i=1}^{n}\left\langle K_{mi}^{X_i}K_{im}^{X_i}\right\rangle_{q(X_i)}\right)\right)^{-1}\right)\right)$$

$$\text{(B.30)}$$

$$= -\frac{d}{2}\text{Tr}\left(\left(K_{mm} + \beta\left(\sum_{i=1}^{n}\left\langle K_{mi}^{X_i}K_{im}^{X_i}\right\rangle_{q(X_i)}\right)\right)^{-1}\beta\text{d}\left(\sum_{i=1}^{n}\left\langle K_{mi}^{X_i}K_{im}^{X_i}\right\rangle_{q(X_i)}\right)\right) \quad \text{(B.31)}$$

$$+ \frac{\beta d}{2}\text{Tr}\left(K_{mm}^{-1}\text{d}\left(\sum_{i=1}^{n}\left\langle K_{mi}^{X_i}K_{im}^{X_i}\right\rangle_{q(X_i)}\right)\right) - \frac{\beta^2}{2}\text{Tr}\left(\left(\sum_{i=1}^{n}\left\langle K_{mi}^{X_i}\right\rangle_{q(X_i)}Y_i\right)\left(\sum_{i=1}^{n}\left\langle K_{mi}^{X_i}\right\rangle_{q(X_i)}Y_i\right)^T\right.$$

$$\text{(B.32)}$$

$$\cdot\left(K_{mm} + \beta\left(\sum_{i=1}^{n}\left\langle K_{mi}^{X_i}K_{im}^{X_i}\right\rangle_{q(X_i)}\right)\right)^{-1}\beta\text{d}\left(\sum_{i=1}^{n}\left\langle K_{mi}^{X_i}K_{im}^{X_i}\right\rangle_{q(X_i)}\right) \quad \text{(B.33)}$$

$$\left.\cdot\left(K_{mm} + \beta\left(\sum_{i=1}^{n}\left\langle K_{mi}^{X_i}K_{im}^{X_i}\right\rangle_{q(X_i)}\right)\right)^{-1}\right) \quad \text{(B.34)}$$

Therefore,

$$\frac{\partial\mathcal{F}}{\partial\left(\sum_{i=1}^{n}\left\langle K_{mi}^{X_i}K_{im}^{X_i}\right\rangle_{q(X_i)}\right)} = -\frac{\beta d}{2}\left(K_{mm} + \beta\left(\sum_{i=1}^{n}\left\langle K_{mi}^{X_i}K_{im}^{X_i}\right\rangle_{q(X_i)}\right)\right)^{-1} + \frac{\beta d}{2}K_{mm}^{-1} \quad \text{(B.35)}$$

$$- \frac{\beta^3}{2}\left(\left(K_{mm} + \beta\left(\sum_{i=1}^{n}\left\langle K_{mi}^{X_i}K_{im}^{X_i}\right\rangle_{q(X_i)}\right)\right)^{-1}\right. \quad \text{(B.36)}$$

$$\cdot\left(\sum_{i=1}^{n}\left\langle K_{mi}^{X_i}\right\rangle_{q(X_i)}Y_i\right)\left(\sum_{i=1}^{n}\left\langle K_{mi}^{X_i}\right\rangle_{q(X_i)}Y_i\right)^T \quad \text{(B.37)}$$

$$\left.\cdot\left(K_{mm} + \beta\left(\sum_{i=1}^{n}\left\langle K_{mi}^{X_i}K_{im}^{X_i}\right\rangle_{q(X_i)}\right)\right)^{-1}\right) \quad \text{(B.38)}$$

**The partial derivative** $\frac{\partial \mathcal{F}}{\partial K_{mm}}$

$$d\mathcal{F}(K_{mm}) = d\left(\frac{d}{2}\log Det(K_{mm}) - \frac{d}{2}\log Det(K_{mm} + \beta\left(\sum_{i=1}^{n}\left\langle K_{mi}^{X_i} K_{im}^{X_i}\right\rangle_{q(X_i)}\right))\right) \tag{B.39}$$

$$+ \frac{\beta d}{2}\text{Tr}(\left(\sum_{i=1}^{n}\left\langle K_{mi}^{X_i} K_{im}^{X_i}\right\rangle_{q(X_i)}\right)K_{mm}^{-1}) \tag{B.40}$$

$$+ \frac{\beta^2}{2}\text{Tr}(\left(\sum_{i=1}^{n}\left\langle K_{mi}^{X_i}\right\rangle_{q(X_i)} Y_i\right)\left(\sum_{i=1}^{n}\left\langle K_{mi}^{X_i}\right\rangle_{q(X_i)} Y_i\right)^T (K_{mm} + \beta\left(\sum_{i=1}^{n}\left\langle K_{mi}^{X_i} K_{im}^{X_i}\right\rangle_{q(X_i)}\right))^{-1})\right) \tag{B.41}$$

$$= \frac{d}{2}\text{Tr}(K_{mm}^{-1} dK_{mm}) - \frac{d}{2}\text{Tr}((K_{mm} + \beta\left(\sum_{i=1}^{n}\left\langle K_{mi}^{X_i} K_{im}^{X_i}\right\rangle_{q(X_i)}\right))^{-1} dK_{mm}) \tag{B.42}$$

$$- \frac{\beta d}{2}\text{Tr}(\left(\sum_{i=1}^{n}\left\langle K_{mi}^{X_i} K_{im}^{X_i}\right\rangle_{q(X_i)}\right)K_{mm}^{-1} dK_{mm} K_{mm}^{-1}) \tag{B.43}$$

$$- \frac{\beta^2}{2}\text{Tr}\left(\left(\sum_{i=1}^{n}\left\langle K_{mi}^{X_i}\right\rangle_{q(X_i)} Y_i\right)\left(\sum_{i=1}^{n}\left\langle K_{mi}^{X_i}\right\rangle_{q(X_i)} Y_i\right)^T (K_{mm} + \beta\left(\sum_{i=1}^{n}\left\langle K_{mi}^{X_i} K_{im}^{X_i}\right\rangle_{q(X_i)}\right))^{-1}\right. \tag{B.44}$$

$$\left. \cdot dK_{mm}(K_{mm} + \beta\left(\sum_{i=1}^{n}\left\langle K_{mi}^{X_i} K_{im}^{X_i}\right\rangle_{q(X_i)}\right))^{-1}\right) \tag{B.45}$$

Therefore,

$$\frac{\partial \mathcal{F}}{\partial K_{mm}} = \frac{d}{2}K_{mm}^{-1} - \frac{d}{2}(K_{mm} + \beta\left(\sum_{i=1}^{n}\left\langle K_{mi}^{X_i} K_{im}^{X_i}\right\rangle_{q(X_i)}\right))^{-1} - \frac{\beta d}{2}K_{mm}^{-1}\left(\sum_{i=1}^{n}\left\langle K_{mi}^{X_i} K_{im}^{X_i}\right\rangle_{q(X_i)}\right)K_{mm}^{-1} \tag{B.46}$$

$$- \frac{\beta^2}{2}(K_{mm} + \beta\left(\sum_{i=1}^{n}\left\langle K_{mi}^{X_i} K_{im}^{X_i}\right\rangle_{q(X_i)}\right))^{-1}\left(\sum_{i=1}^{n}\left\langle K_{mi}^{X_i}\right\rangle_{q(X_i)} Y_i\right)\left(\sum_{i=1}^{n}\left\langle K_{mi}^{X_i}\right\rangle_{q(X_i)} Y_i\right)^T \tag{B.47}$$

$$\cdot (K_{mm} + \beta\left(\sum_{i=1}^{n}\left\langle K_{mi}^{X_i} K_{im}^{X_i}\right\rangle_{q(X_i)}\right))^{-1} \tag{B.48}$$

## B.2 The partial derivatives of the ARD kernel

Here we will look at the partial derivatives $\partial K_{mm}$, $\partial\left\langle K_{ii}^{X_i}\right\rangle_{q(X_i)}$, $\partial\left\langle K_{mi}^{X_i}\right\rangle_{q(X_i)}$, and $\partial\left\langle K_{mi}^{X_i} K_{im}^{X_i}\right\rangle_{q(X_i)}$ with respect to the variables $(Z_{jk})$, $(\mu_i, S_i)_{i \leq N}$, and $\theta = (\sigma_f^2, \alpha_1, ..., \alpha_Q)$.

### B.2.1 Partial derivatives with respect to $Z_{jk}$

**The partial derivative** $\frac{\partial K_{mm}}{\partial Z_{jk}}$

$$\left(\frac{\partial K_{mm}}{\partial Z_{jk}}\right)_{mm'} = \frac{\partial k(Z_m, Z_{m'})}{\partial Z_{jk}} = I(m = j \wedge m' \neq j \vee m \neq j \wedge m' = j)k(Z_m, Z_{m'})(-\alpha_k)(Z_{mk} - Z_{m'k}) \tag{B.49}$$

**The partial derivative** $\frac{\partial\left\langle K_{ii}^{X_i}\right\rangle_{q(X_i)}}{\partial Z_{jk}}$

$$\frac{\partial\left\langle K_{ii}^{X_i}\right\rangle_{q(X_i)}}{\partial Z_{jk}} = 0 \tag{B.50}$$

The partial derivative $\dfrac{\partial \left\langle K_{mi}^{X_i} \right\rangle_{q(X_i)}}{\partial Z_{jk}}$

$$\left( \frac{\partial \left\langle K_{mi}^{X_i} \right\rangle_{q(X_i)}}{\partial Z_{jk}} \right)_m = I(m=j)(\left\langle K_{mi}^{X_i} \right\rangle_{q(X_i)})_m \left( \frac{\alpha_k(\mu_{ik} - Z_{mk})}{\alpha_k S_{ik} + 1} \right) \tag{B.51}$$

Note that we are interested, for the calculation of the lower bound of the log-marginal likelihood, in the derivative of $\left\langle K_{mi}^{X_i} \right\rangle_{q(X_i)} Y_i$ which equals $\dfrac{\partial \left\langle K_{mi}^{X_i} \right\rangle_{q(X_i)}}{\partial Z_{jk}} Y_i$

The partial derivative $\dfrac{\partial \left\langle K_{mi}^{X_i} K_{im}^{X_i} \right\rangle_{q(X_i)}}{\partial Z_{jk}}$

$$\left( \frac{\partial \left\langle K_{mi}^{X_i} K_{im}^{X_i} \right\rangle_{q(X_i)}}{\partial Z_{jk}} \right)_{mm'} = I(m=j)(\left\langle K_{mi}^{X_i} K_{im}^{X_i} \right\rangle_{q(X_i)})_{mm'} \tag{B.52}$$

$$\cdot \left( -\frac{\alpha_k(Z_{mk} - Z_{m'k})}{2} + \frac{\alpha_k(2\mu_{ik} - Z_{mk} - Z_{m'k})}{2(2\alpha_k S_{ik} + 1)} \right) \tag{B.53}$$

### B.2.2 Partial derivatives with respect to $\sigma_f^2$

The partial derivative $\frac{\partial K_{mm}}{\partial \sigma_f^2}$

$$\left( \frac{\partial K_{mm}}{\partial \sigma_f^2} \right)_{mm'} = \frac{k(Z_m, Z_{m'})}{\sigma_f^2} \tag{B.54}$$

The partial derivative $\dfrac{\partial \left\langle K_{ii}^{X_i} \right\rangle_{q(X_i)}}{\partial \sigma_f^2}$

$$\frac{\partial \left\langle K_{ii}^{X_i} \right\rangle_{q(X_i)}}{\partial \sigma_f^2} = 1 \tag{B.55}$$

The partial derivative $\dfrac{\partial \left\langle K_{mi}^{X_i} \right\rangle_{q(X_i)}}{\partial \sigma_f^2}$

$$\left( \frac{\partial \left\langle K_{mi}^{X_i} \right\rangle_{q(X_i)}}{\partial \sigma_f^2} \right)_m = \frac{(\left\langle K_{mi}^{X_i} \right\rangle_{q(X_i)})_m}{\sigma_f^2} \tag{B.56}$$

The partial derivative $\dfrac{\partial \left\langle K_{mi}^{X_i} K_{im}^{X_i} \right\rangle_{q(X_i)}}{\partial \sigma_f^2}$

$$\left( \frac{\partial \left\langle K_{mi}^{X_i} K_{im}^{X_i} \right\rangle_{q(X_i)}}{\partial \sigma_f^2} \right)_{mm'} = 2\frac{(\left\langle K_{mi}^{X_i} K_{im}^{X_i} \right\rangle_{q(X_i)})_{mm'}}{\sigma_f^2} \tag{B.57}$$

### B.2.3 Partial derivatives with respect to $\alpha_q$

Note: in the Python implementation, we have $\alpha_q = \dfrac{1}{l^2}$, therefore $\dfrac{\partial \alpha_q}{\partial l} = -2\dfrac{1}{l^3}$.

The partial derivative $\frac{\partial K_{mm}}{\partial \alpha_q}$

$$\left( \frac{\partial K_{mm}}{\partial \alpha_q} \right)_{mm'} = k(Z_m, Z_{m'})(Z_{mq} - Z_{m'q})^2 \tag{B.58}$$

The partial derivative $\frac{\partial \left\langle K_{ii}^{X_i} \right\rangle_{q(X_i)}}{\partial \alpha_q}$

$$\frac{\partial \left\langle K_{ii}^{X_i} \right\rangle_{q(X_i)}}{\partial \alpha_q} = 0 \tag{B.59}$$

The partial derivative $\frac{\partial \left\langle K_{mi}^{X_i} \right\rangle_{q(X_i)}}{\partial \alpha_q}$

$$\left( \frac{\partial \left\langle K_{mi}^{X_i} \right\rangle_{q(X_i)}}{\partial \alpha_q} \right)_m = -\frac{1}{2} (\left\langle K_{mi}^{X_i} \right\rangle_{q(X_i)})_m \left( \left( \frac{\mu_{iq} - Z_{mq}}{\alpha_q S_{iq} + 1} \right)^2 + \frac{S_{iq}}{\alpha_q S_{iq} + 1} \right) \tag{B.60}$$

The partial derivative $\frac{\partial \left\langle K_{mi}^{X_i} K_{im}^{X_i} \right\rangle_{q(X_i)}}{\partial \alpha_q}$

$$\left( \frac{\partial \left\langle K_{mi}^{X_i} K_{im}^{X_i} \right\rangle_{q(X_i)}}{\partial \alpha_q} \right)_{mm'} = (\left\langle K_{mi}^{X_i} K_{im}^{X_i} \right\rangle_{q(X_i)})_{mm'} \tag{B.61}$$

$$\cdot \left( -\frac{(Z_{mq} - Z_{m'q})^2}{4} - \left( \frac{2\mu_{iq} - Z_{mq} - Z_{m'q}}{2(2\alpha_q S_{iq} + 1)} \right)^2 - \frac{S_{iq}}{2\alpha_q S_{iq} + 1} \right) \tag{B.62}$$

### B.2.4 Partial derivatives with respect to $\mu_{iq}$

The partial derivative $\frac{\partial K_{mm}}{\partial \mu_{iq}}$

$$\frac{\partial K_{mm}}{\partial \mu_{iq}} = 0 \tag{B.63}$$

The partial derivative $\frac{\partial \left\langle K_{ii}^{X_i} \right\rangle_{q(X_i)}}{\partial \mu_{iq}}$

$$\frac{\partial \left\langle K_{ii}^{X_i} \right\rangle_{q(X_i)}}{\partial \mu_{iq}} = 0 \tag{B.64}$$

The partial derivative $\frac{\partial \left\langle K_{mi}^{X_i} \right\rangle_{q(X_i)}}{\partial \mu_{iq}}$

$$\left( \frac{\partial \left\langle K_{mi}^{X_i} \right\rangle_{q(X_i)}}{\partial \mu_{iq}} \right)_m = \left( \left\langle K_{mi}^{X_i} \right\rangle_{q(X_i)} \right)_m \left( -\frac{\alpha_q (\mu_{iq} - Z_{mq})}{\alpha_q S_{iq} + 1} \right) \tag{B.65}$$

The partial derivative $\frac{\partial \left\langle K_{mi}^{X_i} K_{im}^{X_i} \right\rangle_{q(X_i)}}{\partial \mu_{iq}}$

$$\left( \frac{\partial \left\langle K_{mi}^{X_i} K_{im}^{X_i} \right\rangle_{q(X_i)}}{\partial \mu_{iq}} \right)_{mm'} = (\left\langle K_{mi}^{X} K_{im}^{X} \right\rangle_{q(X)})_{mm'} \left( -2 \frac{\alpha_q (2\mu_{iq} - Z_{mq} - Z_{m'q})}{2(2\alpha_q S_{iq} + 1)} \right) \tag{B.66}$$

### B.2.5 Partial derivatives with respect to $S_{iq}$

The partial derivative $\frac{\partial K_{mm}}{\partial S_{iq}}$

$$\frac{\partial K_{mm}}{\partial S_{iq}} = 0 \tag{B.67}$$

**The partial derivative** $\frac{\partial \left\langle K_{ii}^{X_i}\right\rangle_{q(X_i)}}{\partial S_{iq}}$

$$\frac{\partial \left\langle K_{ii}^{X_i}\right\rangle_{q(X_i)}}{\partial S_{iq}} = 0 \tag{B.68}$$

**The partial derivative** $\frac{\partial \left\langle K_{mi}^{X_i}\right\rangle_{q(X_i)}}{\partial S_{iq}}$

$$\left(\frac{\partial \left\langle K_{mi}^{X_i}\right\rangle_{q(X_i)}}{\partial S_{iq}}\right)_m = \left(\left\langle K_{mi}^{X_i}\right\rangle_{q(X_i)}\right)_m \left(\frac{1}{2}\left(\frac{\alpha_q(\mu_{iq}-Z_{mq})}{\alpha_q S_{iq}+1}\right)^2 - \frac{1}{2}\frac{\alpha_q}{\alpha_q S_{iq}+1}\right) \tag{B.69}$$

**The partial derivative** $\frac{\partial \left\langle K_{mi}^{X_i} K_{im}^{X_i}\right\rangle_{q(X_i)}}{\partial S_{iq}}$

$$\left(\frac{\partial \left\langle K_{mi}^{X_i} K_{im}^{X_i}\right\rangle_{q(X_i)}}{\partial S_{iq}}\right)_{mm'} = (\left\langle K_{mi}^{X_i} K_{im}^{X_i}\right\rangle_{q(X_i)})_{mm'}\left(2\left(\frac{\alpha_q(2\mu_{iq}-Z_{mq}-Z_{m'q})}{2(2\alpha_q S_{iq}+1)}\right)^2 - \frac{1}{2}\frac{2\alpha_q}{2\alpha_q S_{iq}+1}\right) \tag{B.70}$$

## B.3 Partial derivatives of $KL(q(X_i)||p(X_i))$

We have $q(X_i) = \mathcal{N}(X_i; \mu_i, S_i)$ and $p(X_i) = \mathcal{N}(X_i; 0, I_d)$. Therefore, the Kullback–Leibler divergence can be evaluated analytically by [wiki "multivariate Gaussian distribution: KL divergence"]

$$KL(q(X_i)||p(X_i)) = \int q(X_i) \log\frac{p(X_i)}{q(X_i)}\mathrm{d}X_i = \frac{1}{2}\left(\sum_{q=1}^{Q}(S_{iq}-\log S_{iq}) + \mu_i^T\mu - Q\right). \tag{B.71}$$

Therefore, we have

$$\frac{\partial KL(q(X_i)||p(X_i))}{\partial(\mu_i)} = \mu_i \tag{B.72}$$

and

$$\frac{\partial KL(q(X_i)||p(X_i))}{\partial(S_{iq})} = \frac{1}{2}\left(1 - \frac{1}{S_{iq}}\right) \tag{B.73}$$

## B.4 Partial derivatives of $\mathcal{F}$ with respect to $\beta$

Lastly, we evaluate the partial derivatives of $\mathcal{F}$ with respect to $\beta$.

$$\frac{\partial \mathcal{F}}{\partial \beta} = \frac{nd}{2}\frac{1}{\beta} - \frac{d}{2}\mathrm{Tr}((\beta\sum_{i=1}^{n}\left\langle K_{mi}^{X_i} K_{im}^{X_i}\right\rangle_{q(X_i)} + K_{mm})^{-1}\left(\sum_{i=1}^{n}\left\langle K_{mi}^{X_i} K_{im}^{X_i}\right\rangle_{q(X_i)}\right)) \tag{B.74}$$

$$- \frac{1}{2}\mathrm{Tr}(Y^TY) - \frac{d}{2}\mathrm{Tr}(\left\langle K_{ii}^{X_i}\right\rangle_{q(X_i)}) + \frac{d}{2}\mathrm{Tr}(K_{mm}^{-1}\left(\sum_{i=1}^{n}\left\langle K_{mi}^{X_i} K_{im}^{X_i}\right\rangle_{q(X_i)}\right)) \tag{B.75}$$

$$+ \beta\mathrm{Tr}(\left(\sum_{i=1}^{n}\left\langle K_{mi}^{X_i}\right\rangle_{q(X_i)} Y_i\right)^T (K_{mm} + \beta\left(\sum_{i=1}^{n}\left\langle K_{mi}^{X_i} K_{im}^{X_i}\right\rangle_{q(X_i)}\right))^{-1}\left(\sum_{i=1}^{n}\left\langle K_{mi}^{X_i}\right\rangle_{q(X_i)} Y_i\right)) \tag{B.76}$$

$$- \frac{\beta^2}{2}\mathrm{Tr}(\left(\sum_{i=1}^{n}\left\langle K_{mi}^{X_i}\right\rangle_{q(X_i)} Y_i\right)^T (K_{mm} + \beta\left(\sum_{i=1}^{n}\left\langle K_{mi}^{X_i} K_{im}^{X_i}\right\rangle_{q(X_i)}\right))^{-1}\left(\sum_{i=1}^{n}\left\langle K_{mi}^{X_i} K_{im}^{X_i}\right\rangle_{q(X_i)}\right) \tag{B.77}$$

$$\cdot (K_{mm} + \beta\left(\sum_{i=1}^{n}\left\langle K_{mi}^{X_i} K_{im}^{X_i}\right\rangle_{q(X_i)}\right))^{-1}\left(\sum_{i=1}^{n}\left\langle K_{mi}^{X_i}\right\rangle_{q(X_i)} Y_i\right)) \tag{B.78}$$

## Footnotes

[1] We follow the definition of matrix normal distribution (Arnold, 1981).

[2] For a full treatment of Gaussian Processes, see Rasmussen & Williams (2006).

[3] Or any other distribution of interest, such as the posterior $p(F|Y, X)$ or any predictive distribution.

[4]A thorough review of the different approaches is given in Quiñonero-Candela & Rasmussen (2005).

[5]Based on conditional Gaussian identities (Bishop, 2006, p. 87)

[6]For more information on distributed architectures see (Dean & Ghemawat, 2008)