[Reviews · NeurIPS 2014]

Submitted by Assigned_Reviewer_1

The authors have come up with a novel way to distribute the optimisation of variational Bayesian sparse GPs (for regression and LVMs). By reformulating Titsias' (2009) variational lower bound, such that the training data become independent given the inducing points, training can be parallelised across nodes on a cluster or network, via a Map-Reduce implementation.

There is of course a bit of overhead, because of the optimisation of global parameters, which is negligible considering the overall speed-up and scaling, and the authors demonstrate that with a simple experiment.

It is nice that the reformulation of the variational lower bound unifies both cases (regression becomes a special case of the LVM when the inputs are fixed).

The paper convincingly demonstrates the benefits of the scheme in terms of balanced computational loads across nodes, and scaling up to millions of training datapoints (a number which they claim have never been attempted in the GP literature).
I do not necessarily see a big improvement in terms of prediction accuracy, but this is because it is not clear what the units of the RMSE are. I suspect they are 'minutes of flight delay', in which case I can appreciate that a marginal improvement of a few minutes does matter to airlines.
Lastly, the authors focus of scalable GPLVMs, by showing significant improvement in the classification accuracy of MNIST data (3%).

Quality and clarity are good. I believe the work is significant to the GP and data mining community.

Other comments:

line 78: 'the optimal form'. Be more precise.

eq.(3.1): Missing an integral symbol.

Tables 1,2: How many inducing points are you using in Dist GP? Also, see my comment on units of the RMSE. (If it is 'min of delay' then Disp GP predicts more accurately by 2-3 minutes.)

Figure 5: Add a more detailed caption. What are the left and right panels showing?

line 395: By 'smaller model' I take it you also mean one using a smaller dataset.

line 393: One possible cause for the unexpected result: SCG optimises the GP marginal likelihood (or var. bound) which is shaped by amount of data. The more the data, the sharper the modes are. Using less data, amounts to an annealing-like behaviour of the likelihood surface, thereby giving you better chances of approaching a good mode.
Summary: This work exploits the conditional independence of data given inducing points in Bayesian variational sparse GPs. Parallelization is then readily feasible by map-reduce. I see this work as a serious step towards large-scale deployment of Bayesian non-parametric models.

Submitted by Assigned_Reviewer_2

This paper builds on previous work on sparse Gaussian processes and latent variable models [1,2] to define an efficient distributed GP algorithm that allows for fast training and inference on large datasets. This is possible due to the way that these models ([1,2]) are constructed, that is, encapsulating an inducing point representation that decouples the data.

This paper is well written, easy to follow, technically sound and addresses a very important problem in the field.

The required factorisations that allow for distributed inference follow almost directly from [2] and [3]; the work in [2] defines the so called Psi statistics that are shown to factorise with respect to the data points and correspond to the quantities denoted by B,C,D in the submitted paper. The work in [3] additionally uses the factorisation in the term y_i * y_i^T, which directly corresponds to the quantity denoted by A in the submitted paper.

However, although the above described previous work [2,3] somehow reduces the originality of the submitted paper, to the best of my knowledge this is the first published work that explicitly puts these pieces together in order to define a parallelisable expression for the objective function. Additionally, the authors embed this expression into a distributed algorithm and analyse its properties. Finally, the experimental evaluation shows that the distributed algorithm is efficient and able to successfully cope with very large datasets.

References
[1] Titsias 2009,"Variational Learning of Inducing Variables in Sparse Gaussian Processes"
[2] Titsias et al. 2010, "Bayesian Gaussian Process Latent Variable Model"
[3] Hensman et al. 2013, "Gaussian Processes for Big Data"
Summary: The theoretical contribution of this paper is quite small given previous work [1,2,3], but the resulting implementation constitutes the first distributed GP algorithm, which gives nice results in real big data.

Submitted by Assigned_Reviewer_15

This paper proposes a re-parametrisation of variational inference for sparse GPs (both for regression and latent variable models) which allows to parallelize the computation across multiple machines. In particular, the authors exploit the fact that conditioned on the inducing inputs, the data decouples and training can be done in parallel in a map-reduce framework. This allows to split the computation of expensive statistics across nodes, while the master node has to simply perform a single matrix inversion (and the size of this matrix doesn't depend on the number of data points or dimensions).

The key idea of the paper is elegant and well explained, the authors investigate all the key aspects of their distributed inference method (including load balacing and robustness to failure). The paper is in general very well written.

minor:
- the URL in footnote 1 is missing
- line 234 "for each output point" -> "for each input point"?
Summary: This is a nice paper that is likely to be appreciated by the NIPS community.
Author Feedback
Author rebuttal: We thank the reviewers for the thorough reading of the paper and picking up on typos, clarifications and other minor comments. We believe the reviewers have understood our paper well. We will go through the comments in order.

Units of the RMSE of the flight dataset (Reviewer_1): the units are indeed in minutes. The marginal improvement is a consequence of the data being very noisy, so the best RMSE would be quite high.

Notation of the inducing targets (Reviewer_2): we chose to stay consistent with the notation of [2].

In response to Reviewer_2's comment about the technical part of our work, we believe it to be broader than what the reviewer suggested. The lower bound presented in our work is algebraically equivalent to the one presented by Titsias et al. [2] (as we mentioned in the paper, sections 1, 4, and the abstract). However our technical contribution was not limited to taking the lower bound and replacing tr(Y Y^T) with tr(Y^T Y) as the reviewer suggests. For example, the Psi_1 statistic does not factorise over n in [2], as it is an N by M matrix. We believe this implies that it was not known to [1,2] that this lower bound factorises and could be parallelised.

We take note of Reviewer_2's suggestions concerning our discussion comparing the inference to [2] and [3]. We discussed how our approach compares to SVIGP [3] in section 2 and we related our method to Titsias [1] and Titsias and Lawrence [2] in section 3. As suggested, we will add a comment on how our method and the SVI method compliment each other, rather than leaving this as an implicit consequence of the disadvantages of SVIGP.

Finally, concerning our experiments, we believe that the experiments we have run are sufficient to support our claims in the paper. We have ensured that the comparison to SVIGP used an experiment setup identical to the one in [3]: we have been in communication with the authors of [3] to obtain the same dataset and pre-processing code that they have used. We will clarify this in the paper.

[1] Titsias 2009,"Variational Learning of Inducing Variables in Sparse Gaussian Processes"
[2] Titsias et al. 2010, "Bayesian Gaussian Process Latent Variable Model"
[3] Hensman et al. 2013, "Gaussian Processes for Big Data"